# Antioxidant Activity, Functional Properties, and Cytoprotective Effects on HepG2 Cells of Tree Peony (*Paeonia suffruticosa* Andr.) Seed Protein Hydrolysate as Influenced by Molecular Weights Fractionation

**DOI:** 10.3390/foods11172592

**Published:** 2022-08-26

**Authors:** Yingying Wang, Yingqiu Li, Chenying Wang, Jinxing He, Haizhen Mo

**Affiliations:** 1School of Food Science & Engineering, Qilu University of Technology (Shandong Academy of Sciences), No. 3501 University Road of Changqing District, Jinan 250353, China; 2School of Food and Biological Engineering, Shaanxi University of Science and Technology, Xi’an 453003, China

**Keywords:** tree peony seed protein hydrolysates, antioxidant activities, functional properties, oxidative stress, cytoprotective effect

## Abstract

In recent years, plant protein hydrolysates have gained increased attention due to their superior antioxidant activity and potential to prevent several chronic diseases associated with oxidative stress. This study aimed to investigate the antioxidant activity, functional properties, and cytoprotective effects of the tree peony seed protein hydrolysate (TPSPH) with different molecular weights (MWs). The antioxidant activities were evaluated by DPPH, hydroxyl radicals scavenging, Fe^2+^ chelating, and inhibition of the β-carotene oxidation abilities. The protective effects and mechanism against oxidative stress were determined using H_2_O_2_-stressed HepG2 cells. MW > 30 kDa of TPSPH showed the highest radical scavenging (DPPH IC_50_ = 0.04, hydroxyl IC_50_ = 0.89 mg/mL) and inhibition of β-carotene oxidation (80.07% at 2.0 mg/mL) activity. Moreover, MW > 30 kDa possessed high hydrophobicity, emulsifying capacity, and abundant antioxidant amino acids (28.22% of hydrophobic amino acids and 8.3% of aromatic amino acids). MW 5–10 kDa exhibited more effective protection against H_2_O_2_-induced HepG2 cells, by reducing reactive oxygen species (ROS), malonaldehyde (MDA), lactate dehydrogenase (LDH), and activating antioxidant enzymes (superoxide dismutase and catalase). These results indicated the potential application of TPSPH as an antioxidant in food and functional foods.

## 1. Introduction

Tree peony (*Paeonia suffruticosa* Andr.) is a kind of famous traditional flower with excellent ornamental characteristics and traditional medicinal values, which has a long history in China [1,2]. The tree peony seed contains amounts of oil and protein nutrient components. In 2011, the Ministry of Health of the People’s Republic of China authorized tree peony seed as a new food resource [1]. In our previous research, we found that the tree peony seed protein (TPSP) was a promising plant protein source that not only had a well-balanced amino acid composition but also some good potentially useful functional characteristics, such as emulsifying and foaming properties. Moreover, the cytotoxicity assay of TPSP showed that 0–400 μg/mL TPSP had no cytotoxicity on human embryonic kidney cells [2].

With in-depth research of plant-derived proteins, enzymatic hydrolysates attracted increasing attention due to their excellent functionality and biological activities, and especially their antioxidant activities [3,4]. Our previous research reported that tree peony seed protein hydrolysate prepared by Alcalase exhibited stronger free radical scavenging, Fe^2+^ chelating activity, and reducing power than other protease hydrolysates and TPSP [5]. Some plant proteins hydrolysates, such as chia, date seed, hemp bran, mung bean, potato, and watermelon seed protein, also possessed good antioxidant properties [6,7,8,9,10,11]. For instance, the date seed protein hydrolysate exhibited a superior capacity to inhibit hydroxyl and peroxyl radicals induced by DNA scission and β-carotene/linoleic acid oxidation. The watermelon seed protein hydrolysate and its ultrafiltration fractions protected HepG2 cells from hydrogen peroxide (H_2_O_2_)-induced oxidative stress by inhibiting lipid peroxidation [7]. There are potential food and pharmaceutical applications for these protein hydrolysates with good antioxidant activities.

The antioxidant activities of protein hydrolysates depend on the protein sources, types of protease, reaction conditions, the degree of hydrolysis, molecular weight, and amino acid compositions, or peptides structure (e.g., side chains and sequences) [4,12]. Different molecular weight (MW) protein hydrolysates exhibit various antioxidant properties. According to Bamdad and Chen [13], Alcalase–barley hordein hydrolysate contained a large peptide fraction (MW > 10 kDa) with high 1,1-diphenyl-2-picrylhydrazyl (DPPH) radical scavenging activity and reducing power as well as small peptides (MW < 1 kDa) with excellent Fe^2+^ chelating capacity. Zhang et al. [14] showed that the MW < 3 kDa fraction of Alcalase-hydrolyzed soybean hydrolysate exhibited the strongest DPPH radical scavenging activity, reducing capacity, and reactive oxygen species (ROS)-quenching ability in human Caco-2 cells, while the MW 5–10 kDa fraction had the best Fe^2+^ chelating capacity. The MW < 1 kDa of the watermelon seed protein hydrolysate was reported to have significant DPPH radical scavenging activity and protect HepG2 cells against oxidative stress induced by H_2_O_2_ when compared to MW > 5 kDa and MW 1–5 kDa [11]. According to Rahimi et al. [10], hydrophobic clusters containing high amounts of Pro, Leu, and Val or bulky side chains with imidazole and benzyl groups are responsible for the high DPPH radical scavenging activity of the MW > 10 kDa fraction of the potato hydrolysate. Low MW hydrolysates or peptides had superior antioxidant effects in vitro and in vivo, which might be attributed to the fact that small peptides have easier steric interactions with reactive radicals, higher hydrophilicity and solubility, or hydrophobic side chain exposure, etc. [3,4,12]. However, there is research literature on antioxidant activity and the protective effects on oxidative stress-induced cell damage of tree peony seed protein hydrolysate (TPSPH) with different molecular weights.

Therefore, to explore the influence of different molecular weights on antioxidant activity and the cytoprotective effect of TPSPH, TPSP hydrolyzed with Alcalase was divided by membrane ultrafiltration into five fractions of different MWs (> 30 kDa, 10–30 kDa, 5–10 kDa, 3–5 kDa, and <3 kDa). The antioxidant capabilities of these five fractions were investigated using the DPPH radical scavenging activity, hydroxyl radical scavenging activity, β-Carotene bleaching assay, and Fe^2+^ chelating activity. The protective effect and mechanism of TPSPH and five fractions of H_2_O_2_-induced oxidative stress in HepG2 cells were explored through ROS level, malonaldehyde (MDA) and lactate dehydrogenase (LDH) levels, as well as the antioxidant enzyme activities of superoxide dismutase (SOD) and catalase (CAT). The amino acid compositions, particle size distribution, surface hydrophobicity, emulsifying and foaming properties, and intrinsic fluorescence spectroscopy were used to determine the physicochemical properties of TPSPH and five fractions.

## 2. Materials and Methods

### 2.1. Materials

Defatted tree peony seed flour was provided by the Huarui Science and Technology Development Co. Ltd. (Heze, Shandong, China). Soybean oil was purchased from a local supermarket in Jinan, Shandong, China. Alcalase 2.4 L (2.0 × 10^5^ U/g) was purchased from Novo Co., Ltd. (Beijing, China). The 1-anilino-8-naphthalene-sulphonate (ANS), DPPH, chloroform, tween-40, β-carotene, linoleic acid, glutathione (GSH), and 2′,7′-dichlorodihydrofluorescein diacetate (DCFH-DA) were purchased from Sigma Co., Ltd. (St. Louis, MO, USA). Dulbecco’s modified Eagle medium (DMEM) and fetal bovine serum (FBS) were purchased from Gibco (Grand Island, NY, USA). Penicillin–streptomycin, 0.25% trypsin-EDTA, and trichloroacetic acid (TCA) were purchased from Beijing Solarbio Science Technology Co., Ltd. (Beijing, China). The 3-(4,5-dimethylthiazol-2-yl)-2,5-diphenylthiazolium bromide (MTT) and LDH assay kits were purchased from Nanjing Jiancheng Bioengineering Institute (Nanjing, China). SOD, MDA, CAT, and Bicinchoninic acid (BCA) assay kits were purchased from Beyotime Institute of Biotechnology (Shanghai, China). All other chemicals and reagents used were of analytical grade.

### 2.2. Preparation of Tree Peony Seed Protein Hydrolysate (TPSPH) and Five Fractions

Freeze-dried TPSP was dispersed in deionized water to obtain a concentration of 20 mg/mL and hydrolyzed at pH 8.0, 60 °C, and 4 h with Alcalase activity of 120 U/g (enzyme unit/substrate weight). After hydrolysis, the enzyme was then deactivated in boiling water for 10 min, followed by centrifugation at 8000× *g* at 4 °C for 20 min. The resulting supernatant was lyophilized and kept at −20 °C for further experiments.

Ultrafiltration of TPSPH was performed using an ultrafiltration stirred cell Amicon 8400 (Millipore Corp., Billerica, MA, USA). TPSPH was sequentially passed through ultrafiltration membranes with MW cut-off of 30, 10, 5, and 3 kDa, respectively. Five fractions with molecular weight of >30 kDa, 10–30 kDa, 5–10 kDa, 3–5 kDa, and <3 kDa were collected, lyophilized, and stored at −20 °C until required for further analysis. The concentration of peptides was determined by using the modified Lowry method with bovine serum albumin as the standard [15]. Peptide concentrations of TPSPH and five fractions were modified for further analysis based on the calculating peptide contents from the prepared standard curve.

### 2.3. Determination of Amino Acid Composition of TPSPH

The amino acid compositions of TPSPH and five fractions were determined as described by Wang et al. [5]. Briefly, TPSPH and five fractions (0.05 g) were digested with 5 mL of 6 M HCl for 24 h at 110 °C under a nitrogen atmosphere. The content of Tryptophan (Trp) was determined by an alkaline hydrolysis method in which 5 M NaOH was used instead of 6 M HCl. After removal of the acid by vacuum evaporation, the hydrolysates were appropriately diluted, filtered and analyzed with an L-8900 amino acids auto-analyzer (L-8900 amino acids auto-analyzer, Hitachi Co. Ltd., Tokyo, Japan).

### 2.4. Determination of Particle Size Distribution of TPSPH

The particle sizes of TPSPH and five fractions were determined using the method described by Wang et al. [5] with some modifications. Prior to taking measurements, each sample was dispersed at a final concentration of 3.0 mg/mL in a 0.01 M sodium phosphate buffer (pH 7.0). Then, the obtained dispersion liquid was measured by a laser particle size analyzer (Zetasizer Nano ZS90, Malvern Instruments Co. Ltd., Malvern, UK) according to dynamic light scattering and laser Doppler.

### 2.5. Determination of Intrinsic Fluorescence Spectroscopy of TPSPH

The intrinsic fluorescence spectroscopy of TPSPH and five fractions was measured using the method of Liu et al. [16] with a fluorescence spectrophotometer (F2700 fluorescence spectrophotometer, Hitachi Co. Ltd., Tokyo, Japan). The samples (0.03 mg/mL) prepared in a 0.01 M phosphate buffer (pH 7.0) were scanned in the wavelength range of 300–400 nm with an excitation wavelength of 280 nm and a slit width of 10 nm.

### 2.6. Determination of Surface Hydrophobicity of TPSPH

The determination of surface hydrophobicity followed the method described by Ai et al. [17] using ANS as the fluorescence probe. ANS binds with exposed hydrophobic regions in some unfolded proteins to produce fluorescence, which is positively correlated with hydrophobicity [5]. Each sample was mixed with sodium phosphate buffer (0.01 M, pH 7.0) into a final concentration of 0.3 mg/mL. Next, twenty microliters of 8 mM ANS was added to 4 mL of sample dispersion. The fluorescence intensity of the obtained mixture was measured using a fluorescence spectrophotometer (F2700 fluorescence spectrophotometer, Hitachi Co. Ltd., Tokyo, Japan) with an excitation wavelength of 375 nm, a scanning wavelength range of 390–650 nm, and a slit width of 5 nm.

### 2.7. Determination of Functional Properties of TPSPH

#### 2.7.1. Emulsifying Properties

The emulsifying activity index (EAI) and the emulsion stability index (ESI) of TPSPH and five fractions were determined using the method of Pearce and Kinsella [18] with minor modifications. TPSPH and five fractions were dispersed (1 mg/mL) in 0.2 M phosphate buffer (pH 7.0) and the dispersion was adjusted to pH 3.0, 5.0, 7.0, or 9.0 with 1.0 M HCl or 1.0 M NaOH. Aliquots (7 mL) of soybean oil and 21 mL of the sample dispersion were homogenized in a homogenizer (FJ-200, Specimen Model Co. Ltd., Shanghai, China) at a speed of 10,000× *g* for 2 min at 25 °C. Aliquots of the homogenized emulsions (50 μL) were immediately taken at 0 min and 10 min of the bottom of the tube and diluted 100 times with 0.1% SDS solution. The diluted solutions were stirred for 5 s and the absorbance of the diluted solution was measured at 500 nm using a UV-1900 spectrophotometer (Shanghai Metash Instrument Co. Ltd., Shanghai, China).
(1)$$\mathrm{EAI}\;(m^{2}/g)=\frac{\left(4.606\times A_{0}\times \mathrm{DF}\right)}{\left(c\times \varphi \times 10000\right)}$$
(2)$$\mathrm{ESI}\;(\min)=\frac{A_{0}}{A_{0}-A_{10}}\Delta t$$
where A_0_ and A_10_ are the absorbance of the diluted emulsion at 0 min and 10 min after homogenization, respectively, DF is the dilution factor (100), c is the protein concentration (g/mL), Δt is the time interval (10 min), and φ is the soybean oil volume fraction of the emulsion (0.25).

#### 2.7.2. Foaming Properties

Foaming capacity (FC) and foam stability (FS) of TPSPH and five fractions were determined according to the method of Gao et al. [2]. Aliquots (30 mL) of 10 mg/mL TPSPH solutions were dispersed in 0.2 M phosphate buffer (pH 7.0) and the pH was adjusted to 3, 5, 7, or 9 with 1.0 M HCl or 1.0 M NaOH. The mixture was homogenized at a speed of 10,000 rpm for 2 min at 25 °C in a homogenizer (FJ-200, Specimen Model Co Ltd., Shanghai, China).
(3)$$\mathrm{FC}\;(\%)=\left(\frac{V_{0}}{V}\right)\times 100$$
(4)$$\mathrm{FS}\;(\%)=\left(\frac{V_{30}}{V_{0}}\right)\times 100$$
where V (mL) is initial volume before homogenization, V_0_ (mL) and V_30_ (mL) are the foam volumes after homogenization at 0 min and 30 min, respectively.

### 2.8. Determination of Antioxidant Activity of TPSPH

#### 2.8.1. DPPH Radical Scavenging Activity

The DPPH radical scavenging activity of TPSPH and five fractions was determined according to a modified method of Wang et al. [5]. The principle of scavenging DPPH radical is based on the absorbance reduction in ethanol solution of DPPH at 517 nm in the presence of antioxidants [4]. A volume of 2 mL of 0.1 mM DPPH solution in 95% methanol was mixed with 2 mL of sample solution (dissolved in distilled water) at different concentrations. The obtained mixture was shaken and left in the dark for 30 min, and its absorbance was measured at 517 nm. As blank and control, ethanol instead of the sample and DPPH solution were used, respectively. The 50% inhibitory concentration (IC_50,_ mg/mL) was defined as the concentration at which the sample scavenged 50% of the radicals [5].
(5)$$\mathrm{DPPH}\;\mathrm{radical}\;\mathrm{scavenging}\;\mathrm{activity}\left(\%\right)=\frac{A_{\mathrm{control}}-A_{\mathrm{sample}}+A_{\mathrm{blank}}}{A_{\mathrm{control}}}\times 100$$

#### 2.8.2. Hydroxyl Radical Scavenging Activity

The hydroxyl radical scavenging activity of TPSPH and five fractions was determined by the method of de Avellar et al. [19] with some modifications. Hydroxyl radical generated by the Fenton reaction can oxidize Fe^2^^+^ to Fe^3^^+^ in 1,10-phenanthroline solution. The principle of scavenging hydroxyl radical is based on decolonization of 1,10-phenanthroline-Fe^2^^+^ at 536 nm in the presence of antioxidants [4]. An aliquot (1.0 mL) of sample dispersion (dissolved in distilled water) was thoroughly mixed with 1.0 mL of 0.75 mM 1,10-phenanthroline, 0.75 mM FeSO_4_ × 7H_2_O, and 0.01% (*v*/*v*) H_2_O_2_. As blank and control, distilled water instead of the H_2_O_2_ and sample solution were used, respectively. The absorbance of the treated mixture solution was measured at 536 nm after incubation at 37 °C for 1.0 h. The hydroxyl radical scavenging rate was calculated using the following equation:(6)$$\mathrm{Hydroxyl}\;\mathrm{radical}\;\mathrm{scavenging}\;\mathrm{activity}\left(\%\right)=\frac{A_{\mathrm{sample}}-A_{\mathrm{blank}}}{A_{\mathrm{control}}-A_{\mathrm{blank}}}\times 100$$

#### 2.8.3. Ferrous Chelating Activity

The ferrous chelating activity of TPSPH and five fractions was measured using the method of Xie et al. [9] with some modifications. An aliquot (1 mL) of sample dispersion (dissolved in distilled water) was added to 2 mL of ferrous chloride (0.05 mM) and 2 mL of ferrozine (0.5 mM). The mixture was vigorously shaken and incubated at room temperature for 10 min. The absorbance of the ferrous iron–ferrozine complex was measured at 562 nm. A control was conducted with distilled water instead of the sample. The ferrous chelating activity was calculated using the following equation:(7)$$\mathrm{Ferrous}\;\mathrm{chelating}\;\mathrm{activity}\left(\%\right)=\left(1-\frac{A_{\mathrm{sample}}}{A_{\mathrm{control}}}\right)\times 100$$

#### 2.8.4. The β-carotene Bleaching Assay

The β-carotene bleaching assay of TPSPH and five fractions was determined according to the method of Oh and Shahidi [20] with some modifications. In β-carotene linoleic acid assay, β-carotene is extremely susceptible to peroxyl free radical-mediated oxidation of linoleic acid, resulting in loss of conjugation and eventual decolorization [7,20]. The principle of β-carotene bleaching assay is based on the absorbance change of β-carotene at 470 nm in the presence of antioxidants [7].

Briefly, β-carotene (2 mg) was dissolved in 10 mL of chloroform, and 1.2 mL of it was transferred into a flask containing 40 mg of linoleic acid and 400 mg of Tween 40 emulsifier. The chloroform was then removed under a nitrogen stream and 100 mL of distilled water was immediately added to the flask. The resulting mixture was stirred vigorously for 30 min to form an emulsion. A blank without β-carotene was also prepared (40 mg of linoleic acid + 400 mg of Tween 40) in the same way. Sample dispersion (2 mg/mL, 0.5 mL) dissolved in distilled water was mixed with 4.5 mL of the above emulsion. A control without sample (β-carotene + 40 mg of linoleic acid + 400 mg of Tween 40) and a mixture of blank (without β-carotene) was also prepared for each sample (sample + 40 mg of linoleic acid + 400 mg of Tween 40). The absorbance was read immediately at 470 nm after the addition of the emulsion. The tubes were then incubated in a shaking water bath at 50 °C, and the absorbance was read at 15 min intervals over a 105 min period. Antioxidant activity of TPSPH and five fractions in oil-in-water emulsion was subsequently calculated using the following equation:(8)$$\mathrm{Antioxidant}\;\mathrm{activity}\;\left(\%\right)=\left(1-\frac{A_{0}-A_{t}}{A_{0}^{o}-A_{t}^{o}}\right)\times 100$$
where A_0_ and A_t_ are the difference between the absorbance of test sample tubes and blank (without β-carotene) tubes at 0 min and 105 min after incubation, respectively, whereas A₀° and A_t_° are the difference between the absorbance of control and blank (without β-carotene) tubes at 0 min and 105 min after incubation, respectively.

### 2.9. Determination of Cytoprotective Effect of TPSPH on Cell Damage Induced by H_2_O_2_

#### 2.9.1. Cell Culture and Cytotoxicity Assay

HepG2 cells were cultured in DMEM medium containing 10% FBS and 1% penicillin–streptomycin at 37 °C in a humidified atmosphere with 5% CO_2_ in the air_._ The medium was refreshed every 2–3 days, and the cells at 80% to 90% confluence were subcultured and dissociated by using a 0.25% trypsin-EDTA solution.

The cytotoxic effect of TPSPH and five fractions on HepG2 cells was measured by MTT assay as previously described, with some modifications [21]. Briefly, HepG2 cells were seeded in 96-well plate at 7 × 10^3^ cells/well and incubated at 37 °C with 5% CO_2_ for 24 h. Then, the medium was removed and replaced with 200 μL of the DMEM medium containing TPSPH or five fractions (final concentration of 0.10 mg/mL) and incubated for a further 12 h. The control wells only received DMEM medium (without samples). Subsequently, 50 μL of MTT solution (1 mg/mL) was added into each well and incubated at 37 °C for another 4 h. After this, the medium from each well was discarded and washed three times with PBS. Finally, 150 μL of DMSO was added to each well to dissolve the formazan precipitate, and the absorbance was measured at 570 nm by using a microplate reader (CMax Plus, Molecular Devices Co Ltd., San Jose, CA, USA). Results were expressed as the percentage of viable cells compared to the control culture.

#### 2.9.2. Cytoprotective Effect

After incubation in a 96-well plate (7 × 10^3^ cells/well) for 24 h, HepG2 cells were pretreated with TPSPH or five fractions (0.10 mg/mL) for 12 h, followed by incubation with H_2_O_2_ (550 µM) for 24 h. Cells treated only with H_2_O_2_ were used as the damage group (oxidative stress). As described in Section 2.9.1, the MTT assay was used to determine the cytoprotective effects of hydrolysates against H_2_O_2_-induced oxidative damage.

#### 2.9.3. Determination of Intracellular ROS Generation in HepG2 Cells

The intracellular ROS in HepG2 cells was evaluated using 2′,7′-dichlorodihydrofluorescein diacetate (DCFH-DA) as a probe according to the method of Xie et al., [22] with slight modifications. DCFH-DA was hydrolyzed to 2′,7′-dichlorodihydrofluorescein (DCFH) by non-specific esterases when it penetrated into cells. Additionally, the presence of ROS could oxidize DCFH to form 2′,7′-dichlorofluorescein (DCF), which was a strong fluorescent product [3].

In brief, HepG2 cells (6 × 10^4^ cells/well, 200 µL/well) were seeded in a 96-well plate, incubated with TPSPH or five fractions (0.10 mg/mL), and incubated with H_2_O_2_ as described above. Afterwards, the cell culture medium was replaced with DCFH-DA fluorescent probe solution (final concentration of 10 μM in cell culture) and incubated at 37 °C for 30 min. After washing twice with PBS buffer to remove extracellular DCFH-DA, the cells were re-suspended into a single cell suspension. The fluorescence intensity was measured with a multi-mode reader (SpectraMax M5, Molecular Devices Co Ltd., San Jose, CA, USA) at an excitation wavelength of 485 nm and an emission wavelength of 525 nm.

#### 2.9.4. Determination of MDA, LDH, SOD, and CAT Levels in HepG2 Cells

HepG2 cells (seeded at 6 × 10^4^ cells/well in a 6-well plate) were incubated with TPSPH or five fractions (0.10 mg/mL) for 12 h and then the cells were treated with or without 550 µM H_2_O_2_ for 24 h. The treated cells were washed twice with ice-cold PBS, lysed using lysis buffer (20 mM Tris, 150 m M NaCl, 1% Triton X-100, and 1 mM EDTA), and centrifuged at 10,000× *g* for 10 min at 4 °C to remove cell debris. Total protein contents of cell supernatants were determined using the BCA Protein Assay Kit (Beyotime Co Ltd., Shanghai, China), per the manufacturer’s instructions. The levels of MDA, SOD, and CAT in cell lysates were determined by using assay kits (Beyotime Co Ltd., Shanghai, China). The LDH leakage was measured by using an assay kit (Nanjing Jiancheng Bioengineering Institute, Nanjing, China). MDA is expressed as nmol/mg prot. LDH is expressed as U/mL protein. SOD and CAT are expressed as U/mg prot.

### 2.10. Statistical Analysis

All experiments were performed in triplicate, and the average value with the standard error was obtained. Origin 8.0 software (Microcal, Los Angeles, CA, USA) was used for all statistical analyses. The data were analyzed by an analysis of variance (ANOVA) using SPSS 19.0 software (SPSS Inc., Chicago, IL, USA). *p* < 0.05 was defined as a significant difference between samples.

## 3. Results and Discussion

### 3.1. Effects of Molecular Weight Fractionation on Amino Acid Compositions of TPSPH

Table 1 shows the compositions and content of the total amino acids of TPSPH and five fractions. The major amino acid residues in TPSPH and five fractions were all Aspartic acid (Asp, 7.23–10.09%), Glutamic acid (Glu, 11.77–23.10 g/100 g), Leucine (Leu, 4.38–7.57 g/100 g), and Arginine (Arg, 4.38–7.13 g/100 g). Compared to TPSPH and other fractions (19.00–28.96%), MW 10–30 kDa, MW 5–10 kDa, and MW < 3 kDa showed high levels of 33.18%, 32.48%, and 32.37% of the acidic amino acids with a negative charge (Asp and Glu), respectively. Hydrophobic amino acids (HAAs; Alanine (Ala), Valine (Val), Isoleucine (Ile), Leucine (Leu), Phenylalanine (Phe), Proline (Pro), Methionine (Met), and Cysteine (Cys)) of the MW > 30 kDa accounted for 28.22% of the total amino acids, which was significantly higher than those of TPSPH (23.37%) and the other fractions (19.15–24.94%) (*p* < 0.05). Compared with the other fractions (4.82–6.19%), MW > 30 kDa of TPSPH also contained high amounts 8.30% of aromatic amino acids (AAAs).

### 3.2. Effects of Molecular Weight Fractionation on Particle Size Distribution of TPSPH

The particle size distributions of TPSPH and five fractions are presented in Figure 1. TPSPH was characterized by a wide distribution of particle sizes between 100 and 10,000 nm, with four peaks in the particle size distributions (PSDs). MW > 30 kDa exhibited three peaks at 78.82–615.10 nm (90.10%), 1718–3091 nm (5.40%), and 4145–6439 nm (4.50%). There were bimodal distributions of MW 10–30 kDa, MW 5–10 kDa, and MW < 3 kDa, which were MW 10–30 kDa (58.57–105.7 nm (13.40%) and 190.10–531.20 nm (88.60%)), MW 5–10 kDa (220.2–342.0 nm (79.0%), and 458.7–531.2 nm (21.0%)), and MW < 3 kDa (68.06–122.4 nm (72.10%) and 164.2–220.2 (27.9%)), respectively. The MW 3–5 kDa only exhibited single-peak particle distribution at 190.1–255 nm. These results indicate that the aggregation phenomenon existed in MW 10–30 kDa, MW 5–10 kDa, MW 3–5 kDa, and MW < 3 kDa due to the action of the chemical bonds that caused small peptides and amino acids to aggregate into fragments or grains.

### 3.3. Effects of Molecular Weight Fractionation on Intrinsic Fluorescence Spectroscopy of TPSPH

The intrinsic fluorescence spectrum of a protein can be used to investigate the polarity of the aromatic amino acid (especially Trp residues) microenvironment and to provide information about changes in tertiary structure [18]. Figure 2 shows the intrinsic fluorescence spectra of TPSPH and five fractions in the range of 320–500 nm. The λmax values of TPSPH and five fractions (337.5–340 nm) greater than 330 nm indicate that the Trp residues of TPSP were exposed [5]. The fluorescence intensities at λmax increased significantly with the increasing molecular weight (*p* < 0.05). The MW > 30 kDa had the highest fluorescence spectra, reaching a maximum fluorescence intensity of 5570, followed by MW 10–30 kDa (5218), while the MW 3 kDa had the lowest fluorescence peak (4486). These results indicate that MW > 30 kDa might have a partially unfolded protein structure with more Trp residues to be exposed to the aqueous environment [16].

### 3.4. Effects of Molecular Weight Fractionation on Surface Hydrophobicity of TPSPH

In general, the surface hydrophobicity is positively related to the fluorescence intensity of ANS when binding to the proteins or hydrolysates [17]. The fluorescence spectra of TPSPH and five fractions are depicted in Figure 3. The fluorescence intensities of MW > 30 kDa (1727) at λmax of 457 nm, 10–30 kDa (477.7) at λmax of 464 nm, MW 5–10 kDa (466.6) at λmax of 463 nm were higher than that of TPSPH (402.6) at λmax of 492.5 nm, indicating that these three fractions had high surface hydrophobicity in comparison to TPSPH. The fluorescence intensities of five fractions increased significantly with the increasing molecular weight (*p* < 0.05), with the fluorescence intensity (1727) of MW > 30 kDa about three to six times higher than the fluorescence intensities of the other low-MW fractions (303.5–477.7). The highest fluorescence intensity of MW > 30 kDa indicated that it had the highest surface hydrophobicity among these fractions. The surface hydrophobicity of bambara groundnut, barley hordein, potato, and rice glutelin protein hydrolysates was similarly found to increase with increasing molecular weight in previous studies [10,13,23,24]. The high surface hydrophobicity of MW > 30 kDa of TPSPH might be attributed to the high molecular weight peptides with partially unfolded protein structures exposing more HAAs/AAAs on the surface [13,24]. These results confirm the results from intrinsic fluorescence spectroscopy (Figure 2).

### 3.5. Effects of Molecular Weight Fractionation on Functional Properties of TPSPH

#### 3.5.1. Emulsifying Properties

The ability of TPSPH and five fractions to form an emulsion could be reflected by the emulsifying activity index (EAI). As shown in Figure 4A, the EAI values (7.63–19.09 m^2^/g) of TPSPH and five fractions at pH 3.0 and 5.0 were lower than their EAI values at pH 7.0 and 9.0 (24.32–192.69 m^2^/g). The lower EAI of TPSPH and five fractions at pH 3.0 and 5.0 might be due to the protein molecules’ aggregation at pH 3.0 and 5.0, close to the isoelectric point of tree peony seed protein, reducing the protein adsorption at the oil–water interface [2]. Singh et al. [25] also reported that rice bran protein hydrolysates with different degrees of hydrolysis had low EAI near their isoelectric point pH 5.0. As the pH increased from 7.0 to 9.0, the EAI values of TPSPH and five fractions increased from 9.95 m^2^/g to 192.69 m^2^/g. The high EAI values of TPSPH and five fractions at pH 7.0 and 9.0 indicated that TPSPH and five fractions had lower protein aggregation in the neutral and alkaline environments, which promoted protein unfolding and thus enhanced their interaction between the oil and water phase [26,27]. The EAI values of MW > 30 kDa increased from 60.63 m^2^/g to 192.69 m^2^/g with an increasing pH of 5.0 to 9.0, which was significantly higher than those of low-MW fractions (9.95–72.24 m^2^/g) (*p* < 0.05). Similarly, high-MW peptide of rice bran protein hydrolysates (23.4–44.5 m^2^/g) also exhibited stronger emulsifying properties than low-MW peptides (17.7–34.4 m^2^/g) at pH 5.0 to 9.0 [25]. It is reported that the EAI of protein/peptide was principally related to surface hydrophobicity and molecular weight [27]. In general, low molecular weight peptides had strong hydrophilic and weak emulsifying properties [26]. The larger molecular weight peptides or more hydrophobic peptides contributed to the stronger EAI of protein/peptide [2]. Therefore, the high emulsifying activity of MW > 30 kDa might be attributed to its high HAAs/AAAs and surface hydrophobicity, increasing the balance between hydrophilicity and hydrophobicity to better wrap or adsorb the oil. The emulsifying capacity (86.04 m^2^/g) of MW > 30 kDa at pH 7.0 was much greater than those of Persian lime seed protein hydrolysates (< 40 m^2^/g) and rice bran protein hydrolysates (28–34.5 m^2^/g), indicating that MW > 30 kDa in neutral and alkaline environments could be used as an alternative to emulsifiers in the food industry [25,28].

The emulsion stability index (ESI) of TPSPH and five fractions at pH 3.0, 5.0, 7.0, and 9.0 is presented in Figure 4B. The ESI of TPSPH decreased as the pH increased, with a maximum ESI of 64.98 min at pH 3.0. The MW 10–30 kDa had a maximum ESI value (66.94 min) at pH 5.0. The maximum ESI values of TPSPH at pH 3.0 and MW 10–30 kDa at 5.0 might be due to the fact that TPSPH and MW 10–30 kDa had isoelectric points at pH 3.0 and 5.0 [2]. The minimal repulsion between protein molecules at the isoelectric point led to the shortest distance between protein molecules, to favor protein adsorption on the oil–water interface, thereby improving emulsion stability [26].

#### 3.5.2. Foaming Properties

The foaming capacity (FC) reflects the ability of TPSPH and five fractions to form foam. As shown in Figure 4C, MW > 30 kDa, MW 3–5 kDa, and MW < 3 kDa all displayed low FC values at pH 3.0 and 5.0 (0.99–25%). As pH increased to 9.0, TPSPH and four fractions had high FC values (117.50–140.63%), with the exception of MW > 30 kDa (22.50%). These results indicate that alkaline environments were more conducive to increase the flexibility, net charge, and solubility of TPSPH and four fractions, leading to a faster diffusion of protein molecules into the air–water interface for foam formation [25]. Among five fractions, MW > 30 kDa had poor foaming capacity (19.51–25%) while MW 10–30 kDa and MW 5–10 kDa displayed high FC values (106.25–148.39%) in all pH ranges. The flexible and balanced hydrophilic/hydrophobic structure of MW 10–30 kDa and MW 5–10 kDa might contribute to their high FC, which might facilitate them to diffuse and adsorb, as well as form foam on the interface [26,27]. The FC values of MW 10–30 kDa (135.48%) and MW 5–10 kDa (131.25%) at pH 7.0 were higher than those of Persian lime seed protein hydrolysates (<80%) and TPSP (about 12%), which demonstrated that MW 10–30 kDa and MW 5–10 kDa had a wider application in food processing such as ice creams, cakes, and meringues in neutral and alkaline environments [2,28].

The foaming stability (FS) of TPSPH and five fractions at different pH levels is presented in Figure 4D. Among the five fractions, MW > 30 kDa exhibited a higher FS (54.0–57.5%) than the other low molecular weight fractions (0.79–46.81%) in the pH range of 3.0 to 9.0. The higher FS of MW > 30 kDa might also be attributed to the higher hydrophobicity of MW > 30 kDa, which enhanced the protein–protein interaction and formed the thicker and stronger film surrounding the air bubbles [26]. Singh et al. [25] also stated that large MW rice bran protein hydrolysates had a better foaming stability than low MW hydrolysates.

### 3.6. Effects of Molecular Weight Fractionation on Antioxidant Activity of TPSPH

#### 3.6.1. DPPH Radical Scavenging Activity

DPPH radical scavenging activity was used to assess the antioxidant effect of protein hydrolysates in hydrophobic systems [4]. As shown in Figure 5A, TPSPH and five fractions exhibited increasing DPPH radical scavenging rates in a dose-dependent manner at 0.02–0.18 mg/mL. At the concentration of 0.18 mg/mL, the DPPH radical scavenging rates of five fractions were 89.37% for MW > 30 kDa, 83.94% for MW 10–30 kDa, 80.56% for MW 5–10 kDa, 63.65% for MW 3–5 kDa, and 54.27% for MW < 3 kDa. MW > 30 kDa (89.37%) and MW 10–30 kDa (83.94%) exhibited considerably strong scavenging activities of DPPH radical in comparison to TPSPH (73.18%) (*p* < 0.05). The lowest IC_50_ value (the highest activity) for MW > 30 kDa was 0.04 mg/mL, followed by 0.11, 0.11, 0.14, 0.14, and 0.17 mg/mL for MW 10–30 kDa, MW 5–10 kDa, MW 3–5 kDa, GSH, and MW < 3 kDa, respectively (Table 2). The IC_50_ 0.04 mg/mL of MW > 30 kDa was much lower than those of the peptide fractions from soybean hydrolysate (IC_50_ = 2.56–8.30 mg/mL) and cottonseed proteins hydrolysate (IC_50_ = 0.49–0.90 mg/mL) [14,29]. These results indicate that MW > 30 kDa of TPSPH had a much stronger DPPH radical scavenging activity than these hydrolysates. Antioxidant hydrolysates generally quench DPPH radicals by transfer reactions of electron transfer and hydrogen atom. The electron transfer reaction is faster and less controlled by diffusion than the hydrogen atom transfer [12]. Therefore, these results indicate that >30 kDa of TPSPH might contain more amino acid groups that could readily donate electrons to DPPH free radicals.

#### 3.6.2. Hydroxyl Radical Scavenging Activity

Hydroxyl radical scavenging activities of TPSPH and five fractions are shown in Figure 5B. The hydroxyl radical scavenging activity of MW > 30 kDa increased from 22.34% to 85.95% when the concentration increased from 0.4 to 2.0 mg/mL. MW > 30 kDa had the lowest IC_50_ value of 0.90 mg/mL among five fractions and GSH (IC_50_ = 1.23–1.86 mg/mL) (Table 2). These data demonstrated that MW > 30 kDa had more potent hydroxyl radical scavenging activity than the low-MW fractions and GSH. The removal of hydroxyl radicals is very important as they could transfer electrons and oxidize organic macromolecules to cause lipid peroxidation and damage to the body [3,21]. The potent hydroxyl radical scavenging activity of MW > 30 kDa further confirmed that MW > 30 kDa might prevent hydroxyl radical-induced lipid peroxidation in food or damage by regulating the biological system [9,11]. Previous studies have established that HAAs (Ala, Val, Ile, Leu, Phe, Tyr, Pro, Met, and Lys) conferred peptides on free radical scavenging capacity due to the abundance of their electrons [5,10,29]. AAAs (Phe, Tyr, and Trp) could scavenge free radicals by direct electron transfer while maintaining their stability through resonance structures [12]. The high radical scavenging activities of the MW > 30 kDa fraction might be attributed to the high amounts of HAAs and AAAs, which were illustrated in the part of amino acid compositions (Table 1). Similar results of superior free radical scavenging activities in high molecular weight fractions have been previously reported for cottonseed protein and barley hordein protein hydrolysates [13,29].

#### 3.6.3. Ferrous Chelating Activity

Fe^2+^ chelating activity was used to study the indirect antioxidant effect of protein hydrolysates. As depicted in Figure 5C, the Fe^2+^ chelating activities of TPSPH and five fractions significantly increased in a dose-dependent manner at 0.05–1.0 mg/mL (*p* < 0.05). The effective chelating capacities of MW 10–30 kDa, MW 5–10 kDa, and MW < 3 kDa for Fe^2+^ (89.67–95.88%) at 1.0 mg/mL were higher than that of TPSPH (61.56%). Moreover, MW 10–30 kDa, MW 5–10 kDa, and MW < 3 kDa had lower IC_50_ values (0.07–0.25 mg/mL) than those of TPSPH, GSH, and the other two fractions (0.43–0.74 mg/mL) (Table 2). The IC_50_ values of MW 10–30 kDa, MW 5–10 kDa, and MW < 3 kDa were lower than those of fractions from soybean hydrolysate (IC_50_ = 0.29–0.45 mg/mL) and cottonseed protein hydrolysate (IC_50_ = 0.94–1.50 mg/mL) [14,29]. These results indicate that MW 10–30 kDa, MW 5–10 kDa, and MW < 3 kDa had a superior capacity for chelating Fe^2+^ to soybean and cottonseed hydrolysates. Peptides containing NCAAs (Asp and Glu) were reported to have a strong ability to chelate metal ions because their charged side-chain groups can form complexes with metal ions [13,23]. In this study, the superior chelating activity of MW 10–30 kDa, MW 5–10 kDa, and MW < 3 kDa fractions of TPSPH might ascribe higher NCAAs amounts, which were illustrated in the part of amino acid compositions (Table 1). These results suggest that MW 10–30 kDa, MW 5–10 kDa, and MW < 3 kDa might have the ability to preserve lipid foods by chelating pro-oxidant metal ions or to reduce damage to cellular components by metal ion-catalyzed oxidation [23,30].

#### 3.6.4. Antioxidant Activity in β-Carotene Bleaching

It has been recognized that high surface-to-volume ratio emulsions are more common than low surface-to-volume bulk lipids which are more common in food and biological systems. Thus, antioxidants need to be added to the oil-in-water emulsion to better fully assess antioxidant activity [31]. The β-carotene linoleic acid assay was commonly used to evaluate the antioxidant activity of antioxidants in oil-in-water emulsions [4,7]. The antioxidant activities of TPSPH and five fractions at a concentration of 2 mg/mL in an oil-in-water emulsion are shown in Figure 5D. The inhibition percentages of β-carotene oxidation at 105 min were 80.07% and 73.60% for the MW > 30 kDa and 10–30 kDa (Figure 5D), respectively, which were significantly higher than TPSPH (57.31%) and the four fractions (48.40–60.42%) (*p* < 0.05). These data indicated that MW > 30 kDa and MW 10–30 kDa exhibited a high inhibitory effect against β-carotene bleaching when compared to TPSPH and low-MW fractions, especially MW > 30 kDa. The highest inhibitory effect against β-carotene bleaching of MW > 30 kDa might be attributed to its strong ability to scavenge free radicals. The high levels of HAAs and surface hydrophobicity in the MW > 30 kDa might play a key role in scavenging radical activity and inhibition of β-carotene bleaching in an oil-in-water emulsion system. In some cases, hydrophobic peptides were more effective than hydrophilic ones in preventing oxidation in oil-in-water emulsions [4,30]. The strong emulsifying ability of hydrophobic peptides could expose more active sites and enhance the interactions between the peptides and the free radicals, thereby improving the ability of peptides to inhibit lipid peroxidation [12]. Moreover, the hydrophobic peptides could form a thick coating around the oil droplets and create a steric barrier, which could block the contact of oil molecules with pro-oxidants in the aqueous phase [4].

All in all, MW > 30 kDa TPSPH exhibited stronger DPPH and hydroxyl radical scavenging activities as well as inhibition of β-carotene bleaching in oil-in-water emulsion systems than those of low-MW fractions and TPSPH. The high hydrophobic character and some antioxidant amino acids (HAAs and AAAs) present in MW > 30 kDa might help to improve the antioxidant activity of MW > 30 kDa. More electrically charged side-chains (NCAAs) were present in the MW 10–30 kDa, MW 5–10 kDa, and MW < 3 kDa, which contributed in their ability to chelate Fe^2+^ ions. Thus, TPSPH and five fractions might have great potential as antioxidants in the emulsion systems of most foods, cosmetics, and biological environments, according to these results.

### 3.7. Effects of Molecular Weight Fractionation on Protective Effects of TPSPH on HepG2 Cells

#### 3.7.1. Cytotoxic Effect

The effect of the survival rates of HepG2 cells treated with various concentrations of TPSPH and five fractions were determined to verify whether TPSPH and five fractions could protect cells from H_2_O_2_-mediated oxidative damage. It was reported that this treatment had no cytotoxic effect on this kind of cell when being over the 80% survival rate of treated HepG2 cells [14]. After being exposed to TPSPH and five fractions at a concentration of 0.07–0.10 mg/mL for 12 h, the viability of HepG2 cells remained above 90% (Figure 6A), indicating that there were no toxic effects on HepG2 cells from these treatments during incubation. Therefore, the aliquots of 0.10 mg/mL samples were used for the subsequent protective concentration of the cells in the oxidative damage model.

#### 3.7.2. Cytoprotective Effect

The cytoprotective effects of TPSPH and five fractions on H_2_O_2_-induced oxidative stress in HepG2 cells at 0.10 mg/mL are shown in Figure 6B. The viability of HepG2 cells (60.61%) was significantly decreased (*p <* 0.05) when exposed to H_2_O_2_ (500 μM) for 24 h as compared with the control cells (100%). The viability of HepG2 cells was improved by the pre-incubation with TPSPH and five fractions enhanced (65.63–85.38%), evidencing that TPSPH and five fractions could protect HepG2 from H_2_O_2_-induced damage. The potent protective effects against H_2_O_2_-induced oxidative damage were as follows: MW 5–10 kDa (85.38%) > MW > 30 kDa (78.20%) > MW 10–30 kDa (75.98%) > MW 3–5 kDa (72.37%) > TPSPH (70.98%) > MW < 3 kDa (65.63%). These results show that TPSPH and five fractions could protect cells from H_2_O_2_-induced damage. In recent years, mung bean protein and soy protein hydrolysates also exhibited certain protective effects against H_2_O_2_-induced damage in the Nctc-1469 and Caco-2 cells [14,22].

To compare and further confirm the protective effects of TPSPH and five fractions on the H_2_O_2_-induced oxidative damage to HepG2 cells, the morphology of the treated cells is shown in Figure 6C. Control HepG2 cells were fusiform-shaped, uniform in size, clear in cell boundaries, and grew adherently in compact monolayers with very little shedding of cells. When the H_2_O_2_-treated cells became round, swollen, irregular in shape and size, and had damaged cell membranes, it showed that H_2_O_2_ caused serious damage to HepG2 cells. With a few irregular, round, swollen cells, the cells treated with TPSPH and five fractions, especially MW 5–10 kDa, restored their spindle shape and clear cell borders. This indicated that TPSPH and five fractions, especially MW 5–10 kDa, reduced cell damage induced by H_2_O_2_.

#### 3.7.3. The ROS Content in HepG2 Cells

The intracellular ROS levels were widely evaluated using the fluorescence intensity of DCF. The effect of TPSPH and five fractions on the ROS contents of H_2_O_2_-induced HepG2 cells is shown in Figure 7A. HepG2 cells treated with H_2_O_2_ had a considerably higher fluorescent intensity (120.0) than the control group (47.87) (*p* < 0.05), indicating that the H_2_O_2_-treated HepG2 cells were under oxidative stress. When pretreated with TPSPH and five fractions, the fluorescence intensities (72.60–113.23) of HepG2 cells were visibly lower than that of the H_2_O_2_-treated group (120.0). The fluorescence intensities of HepG2 cells were in the following order: MW < 3 kDa (113.23), MW 3–5 kDa (111.35), MW 10–30 kDa (104.60), TPSPH (100.58), MW > 30 kDa (91.88), and MW 5–10 kDa (72.60). These results demonstrate that treatment with TPSPH and five fractions could scavenge ROS to protect HepG2 cells. The MW 5–10 kDa of TPSPH displayed the lowest fluorescence intensity, indicating that the MW 5–10 kDa had the strongest capacity to scavenge intracellular ROS among these five fractions. The MW 10–100 kDa and MW 3–10 kDa of the red tilapia scale hydrolysate showed a better intracellular ROS scavenging ability than the MW < 3 kDa, according to studies reported by Sierra et al. [32].

#### 3.7.4. The Content of LDH and MDA in HepG2 Cells

The damage to cells induced by H_2_O_2_ often results in early changes in cell membrane permeability that make the cytoplasmic enzymes leak, such as lactate dehydrogenase (LDH) [33]. Malondialdehye (MDA), a lipid peroxidation product, is formed when the overproduction of ROS oxidizes polyunsaturated fatty acids present in biological membranes [1]. Therefore, LDH and MDA levels were used to assess the degree of membrane damage and lipid peroxidation [34]. Figure 7B,C show MDA and LDH levels in HepG2 cell cultures after treatment with H_2_O_2_ alone, and treatment with TPSPH and five fractions. MDA (26.75 nmol/mg prot) and LDH (0.98 U/mL) contents increased after H_2_O_2_ incubation when compared to the control group (13.90 nmol/mg prot for MDA, 0.64 U/mL for LDH), indicating that H_2_O_2_ induced damage to the HepG2 cells membrane and the occurrence of lipid peroxidation [14,22]. However, the MDA and LDH contents were significantly reduced in TPSPH and five fractions treatments (*p* < 0.05). Among the five fractions, the MW 5–10 kDa of TPSPH exhibited the highest inhibitory effect with the lowest MDA and LDH levels (15.77 nmol/mg prot and 0.68 U/mL), suggesting that the MW 5–10 kDa of TPSPH might be effective in reducing ROS-mediated membrane damage.

#### 3.7.5. Activity of Antioxidant Enzymes in HepG2 Cells

Under normal physiological conditions, SOD and CAT, the primary antioxidant defense systems, could eliminate ROS and protect cells from damage caused by oxidants [14,35]. The highly reactive superoxide anions might be catalyzed by SOD into O_2_ and the less reactive H_2_O_2_. H_2_O_2_ might be converted by CAT to H_2_O and O_2_, preventing the formation of more reactive substances, such as hydroxyl radicals [1]. Figure 7D,E depict the effect of TPSPH and five fractions on the SOD and CAT activities of H_2_O_2_-induced HepG2 cells. After the HepG2 cells were treated with 500 µM H_2_O_2_ for 24 h, the activities of SOD and CAT were 45.93 U/mgprot and 70.61 U/mgprot, respectively. These values were significantly lower than those of the control group, which were 73.68 U/mgprot and 104.29 U/mgprot, respectively (*p* < 0.05). These results indicate that H_2_O_2_ disrupted the SOD and CAT activities in the HepG2 cells. The treatment of TPSPH, MW > 30 kDa, MW 10–30 kDa, and MW 5–10 kDa significantly increased the SOD activities of H_2_O_2_-indued HepG2 cells from 45.93 U/mg prot to 53.47–69.20 U/mg prot (*p* < 0.05), with a high 69.20 U/mg prot for MW 5–10 kDa. Additionally, the CAT activities of H_2_O_2_-induced HepG2 cells were significantly enhanced when treated with MW 5–10 kDa, increasing from 70.61 U/mgprot to 95.17 U/mgprot.

It has been reported that antioxidant peptides in cells could, on the one hand, scavenge intracellular ROS by providing hydrogen atoms or electrons to achieve an antioxidant capacity, and, on the other hand, they could regulate the activity of intracellular antioxidant enzymes to resist oxidative damage in cells to achieve indirect antioxidant effects [3,4]. For instance, the MW < 3 kDa fraction of Alcalase-hydrolyzed soybean protein hydrolysate could effectively protect Caco-2 cells from H_2_O_2_-induced oxidative stress via suppressing intracellular ROS accumulation and stimulating antioxidant enzyme activities [14]. Likewise, the high SOD and CAT activities of the H_2_O_2_-induced HepG2 cells treated with the MW 5–10 kDa of TPSPH implied that the MW 5–10 kDa of TPSPH could effectively protect H_2_O_2_-induced HepG2 cells from oxidative stress, not only by scavenging intracellular free radicals and inhibiting lipid peroxidation but also by enhancing endogenous antioxidant defense systems.

## 4. Conclusions

This study reported on the effects of molecular weight on the antioxidant activity, functional properties, and cytoprotective effects on HepG2 cells of TPSPH. The MW > 30 kDa of TPSPH exhibited a relatively high emulsifying capacity, radicals scavenging activities, and effectively inhibited the oxidation of β-carotene in the emulsifying systems. The high surface hydrophobicity and some antioxidant amino acids (AAAs and HAAs) in the long-chain peptides might be responsible for their high emulsifying capacity and the antioxidant activity of MW > 30 kDa. The MW 10–30 kDa, MW 5–10 kDa, and MW < 3 kDa all exhibited excellent Fe^2+^ chelating activity and foaming capacity, which might be related to their high levels of NCAAs (Asp and Glu). The MW 5–10 kDa of TPSPH exhibited the best protective effect on HepG2 cells against oxidative stress induced by H_2_O_2_ when compared to the high molecular weight > 30 kDa. The MW 5–10 kDa reduced the H_2_O_2_-induced oxidative damage in HepG2 cells by scavenging ROS levels, inhibiting lipid peroxidation, and stimulating cellular antioxidant defense systems (SOD and CAT).

## Figures and Tables

**Figure 1 foods-11-02592-f001:**
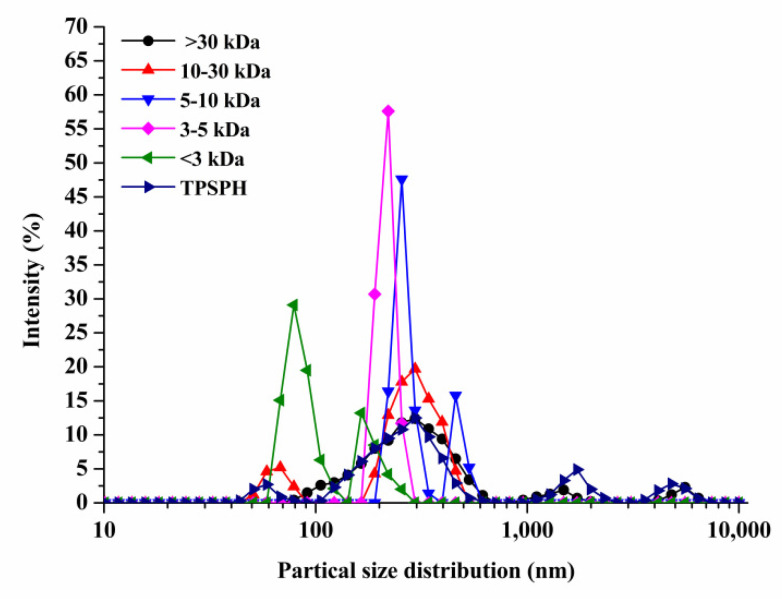
Particle size distribution of TPSPH and five fractions, (●) >30 kDa, (▲) 10–30 kDa, (▼) 5–10 kDa, (♦) 3–5 kDa, (◄) <3 kDa, and (►) TPSPH.

**Figure 2 foods-11-02592-f002:**
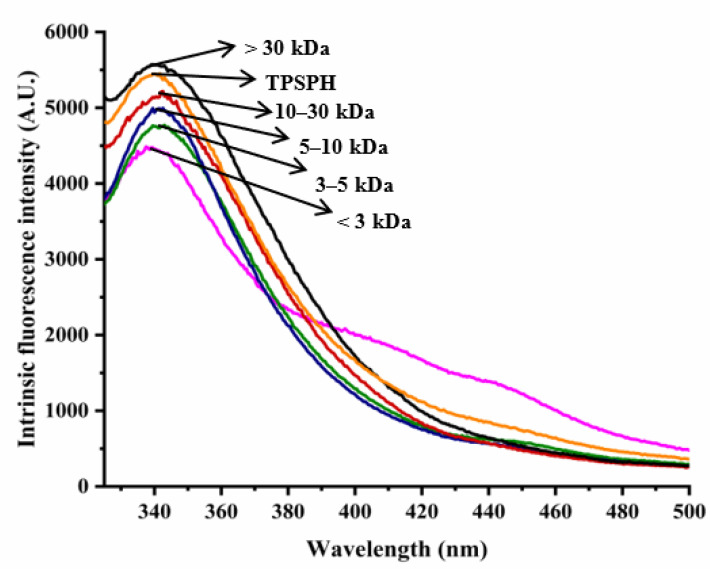
Intrinsic fluorescence spectra of TPSPH and five fractions.

**Figure 3 foods-11-02592-f003:**
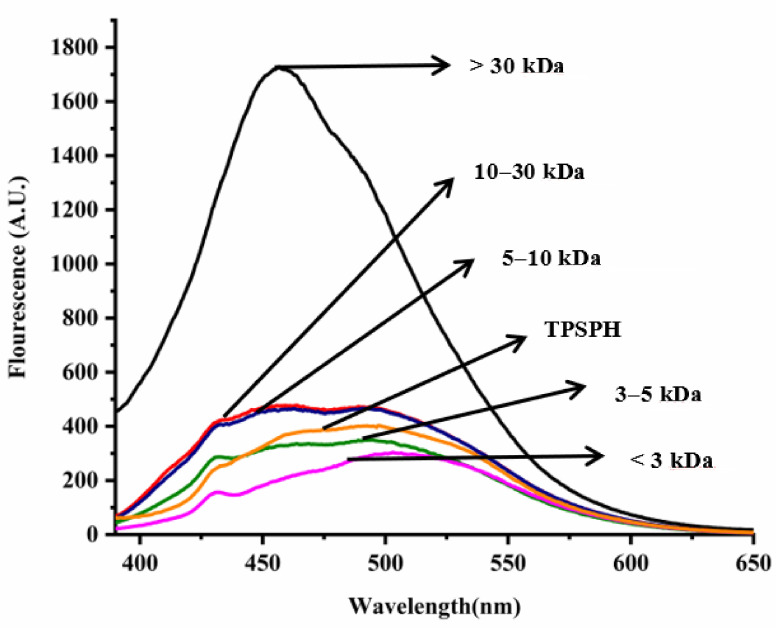
Surface hydrophobicity of TPSPH and five fractions.

**Figure 4 foods-11-02592-f004:**
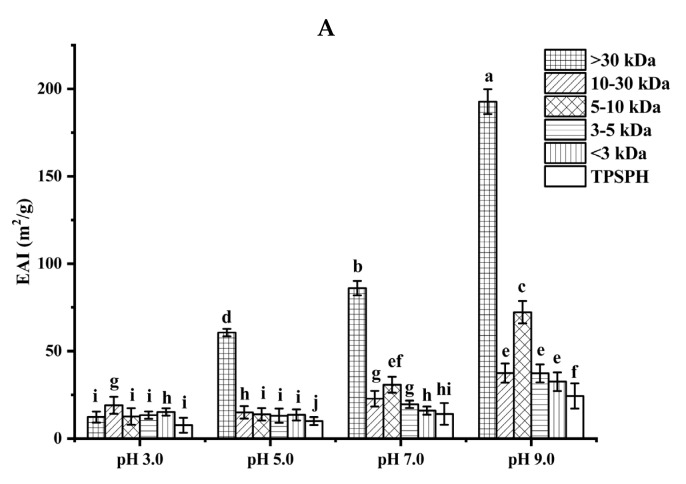
Functional properties of TPSPH and five fractions. (**A**) Emulsifying activity index (EAI). (**B**) Emulsion stability index (ESI). (**C**) Foaming capacity (FC). (**D**) Foaming stability (FS). Different letters indicate significant differences (*p* < 0.05).

**Figure 5 foods-11-02592-f005:**
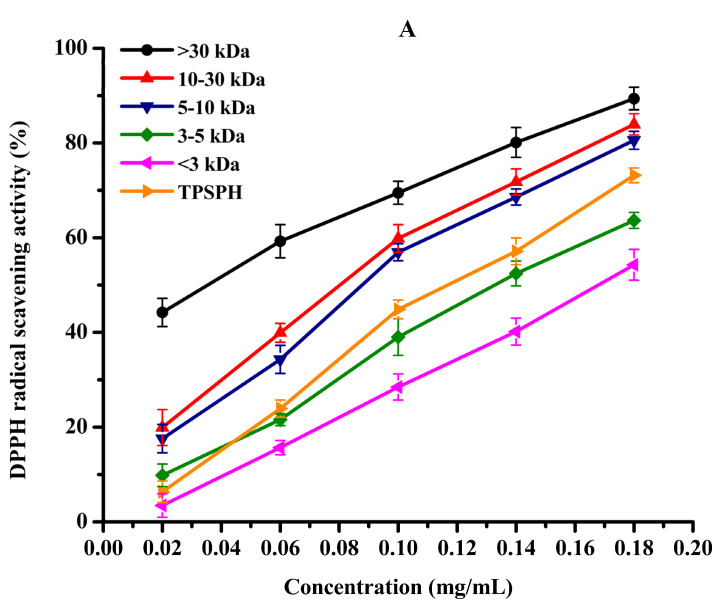
The antioxidant activities of TPSPH and five fractions, (●) >30 kDa, (▲) 10–30 kDa, (▼) 5–10 kDa, (♦) 3–5 kDa, (◄) <3 kDa, and (►) TPSPH. (**A**) DPPH radical scavenging activity. (**B**) Hydroxyl radical scavenging activity. (**C**) Fe^2+^ chelating activity. (**D**) Antioxidant activity against β-carotene bleaching.

**Figure 6 foods-11-02592-f006:**
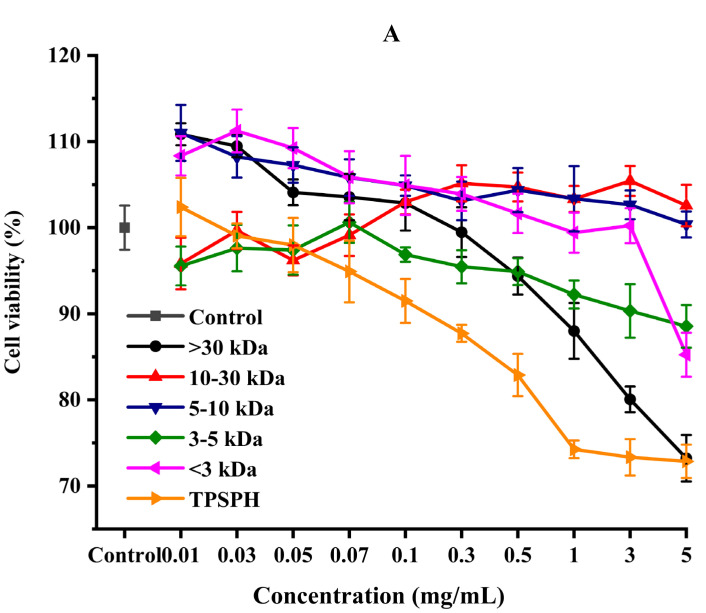
The cytoprotective effects of TPSPH and five fractions on HepG2 cells. (**A**) Cytotoxic effect. (**B**) Protective effects on H_2_O_2_-induced oxidative damage in HepG2 cells. # *p* < 0.05 versus the control group, different letters indicate significant differences between sample groups and the damage groups. (**C**) The morphology of HepG2 cells pre-incubated with TPSPH and its five fractions were observed by microscopy.

**Figure 7 foods-11-02592-f007:**
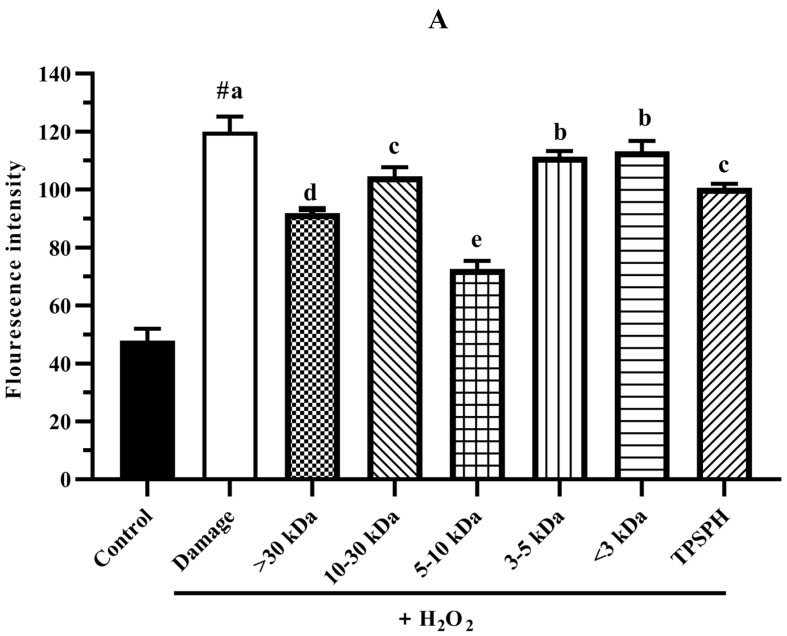
The potential antioxidant mechanisms of TPSPH and five fractions on cell damage induced by H_2_O_2_. (**A**) ROS levels in HepG2 cells (fluorescence intensity of DCF was measured at an excitation wavelength of 485 nm and an emission wavelength of 525 nm). (**B**) MDA levels in HepG2 cells. (**C**) LDH levels in HepG2 cells. (**D**) SOD activity in HepG2 cells. (**E**) CAT activity in HepG2 cells. # *p* < 0.05 versus the control group, different letters indicate significant differences between sample groups and the damage groups.

**Table 1 foods-11-02592-t001:** Amino acid compositions of TPSPH and five fractions.

	Amount (g/100 g) Dry Basis					
	<3 kDa	3–5 kDa	5–10 kDa	10–30 kDa	>30 kDa	TPSPH
Asp	9.46 ± 1.25 ^ab^	7.23 ± 0.85 ^b^	9.48 ± 2.29 ^ab^	10.09 ± 1.76 ^a^	8.32 ± 0.24 ^ab^	8.08 ± 0.17 ^ab^
Thr	2.93 ± 0.24 ^ab^	1.79 ± 0.26 ^d^	2.30 ± 0.35 ^cd^	2.71 ± 0.47 ^ab^	3.26 ± 0.13 ^a^	2.50 ± 0.21 ^bc^
Ser	3.74 ± 0.37 ^b^	0.79 ± 0.08 ^d^	2.98 ± 0.25 ^c^	3.79 ± 0.55 ^b^	5.58 ± 0.12 ^a^	3.58 ± 0.36 ^b^
Glu	22.91 ± 0.89 ^a^	11.77 ± 0.63 ^b^	22.99 ± 1.37 ^a^	23.10 ± 2.75 ^a^	20.09 ± 1.82 ^a^	20.88 ± 1.32 ^a^
Gly	3.94 ± 0.33 ^a^	4.02 ± 0.27 ^a^	3.28 ± 0.22 ^c^	3.47 ± 0.34 ^ab^	4.19 ± 0.22 ^a^	3.67 ± 0.18 ^ab^
Ala	3.51 ± 0.25 ^b^	5.86 ± 0.73 ^a^	2.71 ± 0.62 ^b^	3.06 ± 0.48 ^b^	4.91 ± 0.66 ^a^	3.13 ± 0.44 ^b^
Cys	2.15 ± 0.62 ^b^	0.90 ± 0.28 ^c^	1.10 ± 0.26 ^c^	0.34 ± 0.07 ^d^	0.20 ± 0.05 ^d^	2.91 ± 0.19 ^a^
Val	3.66 ± 0.17 ^bc^	4.27 ± 0.58 ^b^	3.41 ± 0.29 ^c^	3.66 ± 0.36 ^bc^	5.34 ± 0.16 ^a^	5.12 ± 0.37 ^a^
Met	0.93 ± 0.25 ^b^	1.16 ± 0.15 ^ab^	0.79 ± 0.07 ^b^	0.81 ± 0.25 ^b^	1.12 ± 0.11 ^ab^	1.35 ± 0.29 ^a^
Ile	3.51 ± 0.21 ^a^	3.78 ± 0.05 ^a^	2.91 ± 0.18 ^b^	2.69 ± 0.26 ^b^	3.43 ± 0.17 ^a^	3.67 ± 0.31 ^a^
Leu	5.73 ± 0.54 ^b^	5.73 ± 0.24 ^b^	4.38 ± 0.12 ^c^	5.09 ± 0.75 ^bc^	7.57 ± 1.05 ^a^	5.75 ± 0.56 ^b^
Tyr	1.58 ± 0.15 ^cd^	1.59 ± 0.25 ^cd^	1.21 ± 0.23 ^d^	1.68 ± 0.33 ^c^	3.10 ± 0.13 ^b^	3.83 ± 0.25 ^a^
Phe	3.22 ± 0.09 ^c^	3.14 ± 0.05 ^c^	3.51 ± 0.05 ^b^	2.56 ± 0.28 ^d^	3.88 ± 0.15 ^a^	3.62 ± 0.12 ^ab^
Lys	2.20 ± 0.24 ^a^	1.92 ± 0.12 ^ab^	1.59 ± 0.25 ^bc^	1.06 ± 0.15 ^d^	1.65 ± 0.22 ^bc^	1.49 ± 0.16 ^c^
His	1.82 ± 0.08 ^a^	1.35 ± 0.05 ^c^	1.38 ± 0.28 ^c^	1.42 ± 0.17 ^bc^	1.83 ± 0.14 ^a^	1.73 ± 0.23 ^ab^
Arg	7.13 ± 0.46 ^a^	6.86 ± 0.15 ^ab^	4.93 ± 0.86 ^c^	4.04 ± 0.25 ^d^	6.52 ± 0.23 ^ab^	6.12 ± 0.38 ^b^
Trp	1.39 ± 0.21 ^a^	0.84 ± 0.23 ^bc^	1.10 ± 0.11 ^ab^	0.48 ± 0.12 ^c^	1.33 ± 0.25 ^a^	1.30 ± 0.33 ^a^
Pro	5.89 ± 0.15 ^a^	3.30 ± 0.07 ^d^	3.86 ± 0.21 ^c^	3.75 ± 0.14 ^b^	4.54 ± 0.09 ^b^	3.56 ± 0.26 ^cd^
TAAs	84.32 ^a^	66.31 ^c^	72.91 ^b^	74.12 ^b^	86.83 ^a^	82.28 ^a^
AAAs	6.19 ^b^	5.57 ^c^	4.82 ^cd^	5.07 ^c^	8.30 ^a^	8.75 ^a^
BCAAs	12.91 ^bc^	13.77 ^b^	10.70 ^d^	11.43 ^c^	16.33 ^a^	14.54 ^b^
HAAs	24.63 ^b^	24.94 ^b^	19.15 ^c^	19.53 ^c^	28.22 ^a^	23.37 ^b^
PCAAs	11.15 ^a^	10.13 ^ab^	7.90 ^c^	6.52 ^d^	9.99 ^ab^	9.34 ^b^
NCAAs	32.37 ^b^	19.00 ^d^	32.48 ^b^	33.18 ^a^	28.41 ^c^	28.96 ^c^
EAAs	27.22 ^b^	25.02 ^c^	20.59 ^d^	19.96 ^d^	29.79 ^a^	27.74 ^ab^

Asp: Aspartic acid; Thr: Threonine; Ser: Serine; Glu: Glutamic acid; Gly: Glycine; Ala: Alanine; Cys: Cysteine; Val: Valine; Met: Methionine; Ile: Isoleucine; Leu: Leucine; Tyr: Tyrosine; Phe: Phenylalanine; Lys: Lysine; His: Histidine; His: Histidine; Arg: Arginine; Trp: Tryptophan; Pro: Proline. Results are expressed as the mean ± standard deviation (*n* = 3). Different letters in the same row correspond to a significant difference at *p* < 0.05. Aromatic amino acids (AAAs) = Phe, Tyr, and Trp; Branch chain amino acids (BCAAs) = Leu, Ile, and Val; Hydrophobic amino acids (HAAs) = Ala, Val, Ile, Leu, Tyr, Phe, Trp, Pro, Met, and Cys; Positively charged amino acids (PCAAs) = Arg, His, and Lys; Negatively charged amino acids (NCAAs) = Asp and Glu.

**Table 2 foods-11-02592-t002:** The 50% inhibitory concentration (IC_50_, mg/mL) of antioxidant properties of TPSPH and five fractions.

Hydrolysates	IC_50_ (mg/mL)		
	DPPH Radical Scavenging Activity	Hydroxyl Radical Scavenging Activity	Fe^2+^ Chelating Activity
> 30 kDa	0.04 ± 0.01 ^c^	0.90 ± 0.10 ^d^	0.62 ± 0.08 ^b^
10–30 kDa	0.11 ± 0.02 ^b^	1.59 ± 0.08 ^ab^	0.10 ± 0.02 ^e^
5–10 kDa	0.11 ± 0.01 ^b^	1.70 ± 0.17 ^ab^	0.25 ± 0.03 ^d^
3–5 kDa	0.14 ± 0.04 ^ab^	1.81 ± 0.20 ^a^	0.43 ± 0.07 ^c^
< 3 kDa	0.17 ± 0.03 ^a^	1.86 ± 0.12 ^a^	0.07 ± 0.01 ^e^
TPSPH	0.12 ± 0.02 ^b^	1.47 ± 0.31 ^b^	0.74 ± 0.02 ^a^
GSH	0.14 ± 0.03 ^ab^	1.23 ± 0.16 ^c^	0.46 ± 0.04 ^c^

Different letters in the same column indicate significant difference (*p* < 0.05). Results are expressed as the mean ± standard deviation (*n* = 3).

## Data Availability

The data presented in this study are available on request from the corresponding author.

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
