# Peer review of "Antioxidant Activity, Functional Properties, and Cytoprotective Effects on HepG2 Cells of Tree Peony (Paeonia suffruticosa Andr.) Seed Protein Hydrolysate as Influenced by Molecular Weights Fractionation"

_foods, 2022, doi:10.3390/foods11172592_

Round 1
Reviewer 1 Report
The authors studied in detail the antioxidant properties of different fractions of the protein hydrolysate of tree peony seeds, as well as effect of these fractions on the survival of HepG2 cells treated with hydrogen peroxide. Fractions have been found that have antioxidant activity and increase cell survival.
The main problem of this work is that the peptide fractions were obtained by ultrafiltration of the protein hydrolysate, but the peptide composition of the obtained fractions was not studied. After ultrafiltration, high-molecular-weight fractions may contain significant amount of low-molecular-weight peptides due-to the interactions between the peptides. To determine the peptide composition of the fractions, they should be analyzed by tricine SDS-PAGE.
Other comments:
1) Abbreviation list should be added.
2) Methods (sections 2.5.1 – 2.5.4): It should be indicated (briefly) which reactions were used to measure the antioxidant activities and which products were recorded at the indicated wavelengths.
4) Fig. 4A requires a more detailed caption so that one can understand what fluorescence was measured (excitation, emission) without searching for this information in the text.
5) Line 316 and 722-723: DCFH-DA, DCFH and DCF needs to be decrypted and included in abbreviation list.
6) Some phrases are difficult to read (for example, lines 518-523 and 527-532) and it would be good to edit them.
7) Line 687: 500 mM (misprint).
Author Response
A point-to-point response for revised manuscript # foods-1790850
Title: Antioxidant activity and functional properties of tree peony (Paeonia suffruticosa Andr.) seed protein hydrolysate and its cytoprotective effects on HepG2 cells
Authors: Ying-Qiu Li et al.
Submitted to: Foods.
The manuscript has been revised according to the comments of reviewers. A point-to-point response for revised manuscript was as followed:
Q1: The main problem of this work is that the peptide fractions were obtained by ultrafiltration of the protein hydrolysate, but the peptide composition of the obtained fractions was not studied. After ultrafiltration, high-molecular-weight fractions may contain significant amount of low-molecular-weight peptides due-to the interactions between the peptides. To determine the peptide composition of the fractions, they should be analyzed by Tricine SDS-PAGE.
A: According to the reviewer’s comments, the Tricine SDS-PAGEs of TPSPH and five fractions were performed and the results were described in the revised manuscript.
Q2: Abbreviation list should be added.
A: According to the reviewer’s comments, abbreviation list was added in the end of the revised manuscript.
Q3: Methods (sections 2.5.1-2.5.4): It should be indicated (briefly) which reactions were used to measure the antioxidant activities and which products were recorded at the indicated wavelengths.
A: According to the reviewer’s comments, the reactions of antioxidant activities and products recorded at the indicated wavelengths were indicated in sections 2.9.1-2.9.4.
Q4: Fig. 4A requires a more detailed caption so that one can understand what fluorescence was measured (excitation, emission) without searching for this information in the text.
A: According to the reviewer’s comments, the excitation and emission wavelengths for fluorescence measurements were described in the caption of Fig. 8A.
Q5: Line 316 and 722-723: DCFH-DA, DCFH and DCF needs to be decrypted and included in abbreviation list.
A: According to the reviewer’s comments, DCFH-DA, DCFH and DCF were interpreted and were added in abbreviation list.
Q6: Some phrases are difficult to read (for example, lines 518-523 and 527-532) and it would be good to edit them.
A: According to the reviewer’s comments, some phrases were revised in the revised manuscript.
Q7: Line 687: 500 mM (misprint).
A: According to the reviewer’s comments, “500 mM” was revised to “500 μM” in the revised manuscript.

Reviewer 2 Report
Comments to the Authors
The manuscript entitled “Antioxidant activity and functional properties of tree peony (Paeonia suffruticosa Andr.) seed protein hydrolysate and its cytoprotective effects on HepG2 cells” presents interesting read. But I have the following concerns and observations for further improvement of the manuscript.
The title of the manuscript should be revised to “Antioxidant activity, functional properties and cytoprotective effects on HepG2 cells of tree peony (Paeonia suffruticosa Andr.) seed protein hydrolysate as influenced by molecular weights fractionation”
The introduction section of the manuscript is intended to frame the study against the main objectives. However, the introduction presents flaws in terms of sequence and logic in the presentation of topics in each paragraph, repetition in the construction of the sentences. I recommend a rewrite accompanied by a review of the text that should contain the ideas that the authors want the reader to capture when reading the manuscript. For example, tree peony and its importance should be introduced from the beginning, and the need/reasons for obtaining hydrolysates from the seeds should be clearly articulated. Thereafter, the authors can discuss the health promoting benefits of plant protein hydrolysates. This should be followed by problem statement of functionality of hydrolysates being dependent on molecular weight distributions of hydrolysates with appropriate citations, and why the study is important.
The authors have also failed to fully articulate and convince me of the novelty of the current study. The authors made reference to their previous study:
Wang, Y. Y., Wang, C. Y., Wang, S. T., Li, Y. Q., Mo, H. Z., & He, J. X. Physicochemical properties and antioxidant activities of tree peony (Paeonia suffruticosa Andr.) seed protein hydrolysates obtained with different proteases. Food Chem., 2021, 345, 128765. https://doi.org/10.1016/j.foodchem.2020.128765.
How is the current study different from this study? In lines 105-106, it was stated: “Yet, there was no literature on antioxidant activity and functional properties of tree peony seed protein hydrolysates (TPSPH) with different molecular weight”, but in the referred article, it was stated: “Alcalase hydrolysates possessed the highest degree of hydrolysis (27.97%) and lowest molecular weight (< 13 kDa)”. It seems molecular weight was also studied in the referred publication.
The authors have stated in lines 57-59 that the antioxidant activity of protein hydrolysates depends on amino acid composition, molecular weights and structure of the resulting peptides. Basically, this is in reference to structural conformation change or modification of proteins/peptides during hydrolysis and their particle size. Amino acids evaluate primary structure modification and surface hydrophobicity evaluates tertiary structure modifications in protein structure. The results and discussions on amino acids and surface hydrophobicity should be presented separately from the functional properties and should be discussed in reference/relation to the structural conformation changes of the proteins/peptides. Fluorescence emission intensity investigates the environment of aromatic amino acid residues like tyrosine (Tyr), phenylalanine (Phe) and tryptophan (Trp), which generate endogenous fluorescence at certain excitation wavelengths. What is the impact of the change in aromatic amino acid residues in terms of structure? Refer to the following publications:
LWT-Food Science and Technology: Hemker et al. (2020). Effects of pressure-assisted enzymatic hydrolysis on functional and bioactive properties of tilapia (Oreochromis niloticus) by-product protein hydrolysates. https://doi.org/10.1016/j.lwt.2019.109003
Ultrasonics Sonochemistry: Esua et al. (2022). Functional and bioactive properties of Larimichthys polyactis protein hydrolysates as influenced by plasma functionalized water-ultrasound hybrid treatments and enzyme types. https://doi.org/10.1016/j.ultsonch.2022.106023
Ultrasonics Sonochemistry: Li et al. (2020). Impact of combined ultrasound-microwave treatment on structural and functional properties of golden threadfin bream (Nemipterus virgatus) myofibrillar proteins and hydrolysates. https://doi.org/10.1016/j.ultsonch.2020.105063
The particle size distribution of the hydrolysates should be performed for further confirmation of the results of emulsifying and foaming properties and the surface active and antioxidant properties of TPSPH should have been compared with standards.
In many parts of the manuscript, language is a problem. For example, Lines 13 and 44: “In recent years, plant proteins hydrolysates have captured...” Replace captured with a more appropriate word. Line 49: “…have demonstrated to possess good antioxidant properties” Line 862: “These results open the door for future studies” Revise with appropriate words.
These are just a few and the entire manuscript should be revised for grammar and redundant words. Review all sub-headings and remove ‘five fractions” from the titles. The subheadings for the results and discussions should be revised to read: Effects of molecular weight fractionation on…. of TPSPH
The results and discussions section is very descriptive and needs improvement with proper comparison with literature, especially section 3.3. The results should be discussed in relations with variations in molecular weights from fractionation, and the practicality of the findings must be discussed.
Author Response
A point-to-point response for revised manuscript # foods-1790850
Title: Antioxidant activity and functional properties of tree peony (Paeonia suffruticosa Andr.) seed protein hydrolysate and its cytoprotective effects on HepG2 cells
Authors: Ying-Qiu Li et al.
Submitted to: Foods.
The manuscript has been revised according to the comments of reviewers. A point-to-point response for revised manuscript was as followed:
Q1: The title of the manuscript should be revised to “Antioxidant activity, functional properties and cytoprotective effects on HepG2 cells of tree peony (Paeonia suffruticosa Andr.) seed protein hydrolysate as influenced by molecular weights fractionation”
A: According to the reviewer’s comments, the title of the manuscript was revised in the revised manuscript.
Q2: The introduction section of the manuscript is intended to frame the study against the main objectives. However, the introduction presents flaws in terms of sequence and logic in the presentation of topics in each paragraph, repetition in the construction of the sentences. I recommend a rewrite accompanied by a review of the text that should contain the ideas that the authors want the reader to capture when reading the manuscript. For example, tree peony and its importance should be introduced from the beginning, and the need/reasons for obtaining hydrolysates from the seeds should be clearly articulated. Thereafter, the authors can discuss the health promoting benefits of plant protein hydrolysates. This should be followed by problem statement of functionality of hydrolysates being dependent on molecular weight distributions of hydrolysates with appropriate citations, and why the study is important.
A: According to the reviewer’s comments, the introduction about this manuscript was rewritten.
Q3: The authors have also failed to fully articulate and convince me of the novelty of the current study. The authors made reference to their previous study:
Wang, Y. Y., Wang, C. Y., Wang, S. T., Li, Y. Q., Mo, H. Z., & He, J. X. Physicochemical properties and antioxidant activities of tree peony (Paeonia suffruticosa Andr.) seed protein hydrolysates obtained with different proteases. Food Chem., 2021, 345, 128765. https://doi.org/10.1016/j.foodchem.2020.128765.
How is the current study different from this study? In lines 105-106, it was stated: “Yet, there was no literature on antioxidant activity and functional properties of tree peony seed protein hydrolysates (TPSPH) with different molecular weight”, but in the referred article, it was stated: “Alcalase hydrolysates possessed the highest degree of hydrolysis (27.97%) and lowest molecular weight (< 13 kDa)”. It seems molecular weight was also studied in the referred publication.
A: The aim of our previous study was to investigate the physicochemical and antioxidant properties of the end product of tree peony seed protein (TPSP) enzymatic hydrolysis via five enzymes (alcalase, neutrase, papain, protamex and flavourzyme). Previous results were designed to demonstrate that different proteases had specific active sites and the number of cleavage sites located in the protein chain, so there were differences in the physicochemical and antioxidant properties of the resulting TPSP hydrolysates. Our present study was aimed to further investigate antioxidant activity, functional properties and cytoprotective effects on HepG2 cells of alcalase-hydrolyzed tree peony seed protein hydrolysates as influenced by molecular weights fractionation. This pensent study was performed based on previous findings, which alcalase hydrolysate displayed the highest radical scavenging, Fe2+ chelating activity, and reducing power. Different molecular weight fractions of hydrolysates might exhibit different physicochemical properties and antioxidant activity due to the different lengths of peptide chains they contain. However, our previous study aimed to investigate the molecular weight of TPSP enzymatic hydrolysate. So the novelty of the current study was different from that of our previous study.
Q4: The authors have stated in lines 57-59 that the antioxidant activity of protein hydrolysates depends on amino acid composition, molecular weights and structure of the resulting peptides. Basically, this is in reference to structural conformation change or modification of proteins/peptides during hydrolysis and their particle size. Amino acids evaluate primary structure modification and surface hydrophobicity evaluates tertiary structure modifications in protein structure. The results and discussions on amino acids and surface hydrophobicity should be presented separately from the functional properties and should be discussed in reference/relation to the structural conformation changes of the proteins/peptides.
A: According to the reviewer’s comments, the results and discussions on amino acids and surface hydrophobicity were presented separately from the functional properties in the revised manuscript.
Q5: Fluorescence emission intensity investigates the environment of aromatic amino acid residues like tyrosine (Tyr), phenylalanine (Phe) and tryptophan (Trp), which generate endogenous fluorescence at certain excitation wavelengths. What is the impact of the change in aromatic amino acid residues in terms of structure? Refer to the following publications:
LWT-Food Science and Technology: Hemker et al. (2020). Effects of pressure-assisted enzymatic hydrolysis on functional and bioactive properties of tilapia (Oreochromis niloticus) by-product protein hydrolysates. https://doi.org/10.1016/j.lwt.2019.109003
Ultrasonics Sonochemistry: Esua et al. (2022). Functional and bioactive properties of Larimichthys polyactis protein hydrolysates as influenced by plasma functionalized water-ultrasound hybrid treatments and enzyme types. https://doi.org/10.1016/j.ultsonch.2022.106023
Ultrasonics Sonochemistry: Li et al. (2020). Impact of combined ultrasound-microwave treatment on structural and functional properties of golden threadfin bream (Nemipterus virgatus) myofibrillar proteins and hydrolysates. https://doi.org/10.1016/j.ultsonch.2020.105063
A: According to the reviewer’s comments, the intrinsic fluorescence spectroscopy of TPSPH and five fractions was measured and the results were described in the revised manuscript. Results showed that MW >30 kDa might have a partially unfolded protein structures with more Trp residues to be exposed to the aqueous environment, which was in agreement with the data of surface hydrophobicity.
Q6: The particle size distribution of the hydrolysates should be performed for further confirmation of the results of emulsifying and foaming properties and the surface active and antioxidant properties of TPSPH should have been compared with standards.
A: According to the reviewer’s comments, the particle size distribution of the hydrolysates was performed and the results described in the revised manuscript. And the antioxidant properties of TPSPH were compared with GSH.
Q7: In many parts of the manuscript, language is a problem.
For example, Lines 13 and 44: “In recent years, plant proteins hydrolysates have captured...” Replace captured with a more appropriate word.
Line 49: “…have demonstrated to possess good antioxidant properties”.
Line 862: “These results open the door for future studies”.
Revise with appropriate words. These are just a few and the entire manuscript should be revised for grammar and redundant words.
A: According to the reviewer’s comments, we have carefully revised for grammar and redundant words in the entire manuscript.
Q8: Review all sub-headings and remove ‘five fractions” from the titles. The subheadings for the results and discussions should be revised to read: Effects of molecular weight fractionation on…. of TPSPH.
A: According to the reviewer’s comments, all sub-headings were removed ‘five fractions” from the titles. And subheadings for the results and discussions were revised to “Effects of molecular weight fractionation on…. of TPSPH”.
Q9: The results and discussions section is very descriptive and needs improvement with proper comparison with literature, especially section 3.3. The results should be discussed in relations with variations in molecular weights from fractionation, and the practicality of the findings must be discussed.
A: According to the reviewer’s comments, the results and discussions section were revised with proper comparison with literature, especially section 3.8.

Reviewer 3 Report
The manuscript “Antioxidant activity and functional properties of 2 tree peony (Paeonia suffruticosa Andr.) seed 3 protein hydrolysate and its cytoprotective effects 4 on HepG2 cells “is a novel study that assess the possible use of TPSPH as a potential antioxidant in food and functional foods. This is an interesting study and the authors have for the first time screened tree peony seed protein hydrolysate (TPSPH) against DPPH, hydroxyl radicals scavenging, Fe2+ chelating. The author has also investigated inhibition of β-carotene oxidation by TPSPH. Similarly, the protective effects and mechanism against oxidative stress were determined using H2O2- stressed HepG2 cells. In addition, the antioxidant activity which is one of the propounding factor in degenerative disorders. Overall, the manuscript is well written and structured, and is capable for publishing in this journal.
However, there are some typographical and grammatical errors throughout the manuscript that should be corrected accordingly before publication.
Author Response
A point-to-point response for revised manuscript # foods-1790850
Title: Antioxidant activity and functional properties of tree peony (Paeonia suffruticosa Andr.) seed protein hydrolysate and its cytoprotective effects on HepG2 cells
Authors: Ying-Qiu Li et al.
Submitted to: Foods.
The manuscript has been revised according to the comments of reviewers. A point-to-point response for revised manuscript was as followed:
Q1: However, there are some typographical and grammatical errors throughout the manuscript that should be corrected accordingly before publication.
A: According to the reviewer’s comments, we have carefully revised some typographical and grammatical errors throughout the manuscript.

Reviewer 4 Report
The study aims to investigate the antioxidant activity, functional properties and cytoprotective effects of tree peony seed protein hydrolysate (TPSPH) and its fractions of different molecular weight (MW) range. It supplements the study field with valuable information; the study is novel; however, numerous corrections of manuscript must be made.
Abstract
The abstract contains a lot of abbreviations, most of them are clear for the readers from the field; however, HAAs and AAAs are both called antioxidant amino acids (line 26), what can be wrongly understood as full form of AAAs abbreviation.
Sentences with ‘exhibited more effectively protect effects’ (line 27), and ‘activating the activities’ (line 29) should be changed.
Introduction
‘…challenge problem…’ must be corrected in the first sentence.
It would be clearer for the reader if the explanations of ROS effect in foods and body were not fused in every sentence of the first paragraph but explained separately.
Lines 62-67. The sentence needs a revision (fractionation…exhibited…activities, …scavenge of…).
Lines 68, 71. Abbreviations WSPHs-I and WSPHs are not given in full form, therefore it is difficult to understand the difference between them.
Line 78. ‘…good bioavailable…’ must be corrected.
Lines 96-99. Authors claim that ‘Tree peony (Paeonia suffruticosa Andr.) seed protein is a good source of protein with favourable amino acid profiles and desirable functional properties, such as emulsifying, foaming, water absorption and oil absorption properties’, but do not explain for which purpose these properties are favourable (as a food, food additive, preservative, drug/supplement etc.). Also, some data on the possible toxicity (or absence of toxicity) of tree peony seeds should be mentioned.
Line 111. ‘…’TPSPH hydrolyzed with alcalase…’ must be corrected, since hydrolysate (TPSPH) wasn’t hydrolyzed again.
Materials and Methods
The used sample solvent is missing in Sections 2.4.2, 2.5.1-4 .
Section 2.2. There is no information about mixing during hydrolysis procedure.
Section 2.4.1. There is no explanation how the surface hydrophobicity is determined from fluorescence spectra.
Section 2.4.2. There is ‘emulsion activity’ in the section title and ‘emulsifying activity’ in the text which is not the same. It is unclear how the pH was adjusted in the two-phase liquid system and what instrument was used for homogenization (centrifuge?).
The equal sign must not be used instead of a dash in equation parameters explanation.
Section 2.4.3.
The sentence in lines 213-214 is not full.
It should be noted that initial volume V is before homogenization, and V0 and V30 – after.
Section 2.5.3
Line 248. ‘…activity was conducted…’ must be corrected.
Section 2.5.4.
Lines 278-280. What does ‘corrected absorbance’ mean?
Section 2.6.1.
Line 288. Cells were not subcultured in a 0.25% trypsin-EDTA solution but dissociated/detached using this solution.
Section 2.6.2.
A cell damage, induced by H2O2, cannot be called ‘stimulation’ (also further in the manuscript).
Section 2.6.3.
Why was ROS generation measured in suspension but not in monolayer? How the resuspension procedure was performed (detachment solution, centrifugation, volumes used etc.)?
Section 2.6.4.
Line 332. ‘…were added with…’ must be corrected.
The composition of lysis buffer is not given. Was it the same for all measurements described in the section?
The description of LDH leakage measurement is missing.
Results and discussion
Section 3.1.1.
The content of amino acids is given in percent in the text but in g/100 g dry basis in Table 1. The units should be unified.
Statistical significance marking is missing in the data of grouped amino acids in Table 1.
Section 3.1.2
Lines 384-385. ‘…fluorescence spectrum excited by binding ANS to hydrophobic groups…’ is not correct since the spectrum cannot be excited.
Line 388. ‘Fluorescence emission spectra…’ must be changed to ‘fluorescence spectra’.
Line 388. The wavelength at which the fluorescence intensity was measured is not given.
Fig. 1A y axis name is incorrect. Fig. 1 A-E should be individual pictures, not subpanels.
Section 3.1.3.
Lines 412-415. Authors claim, that TPSPH and five fractions at pH 3.0 and 5.0 exhibited lower EAI values, except for MW >30 kDa at pH 5.0, however, it is unclear compared to what? All EAI values, including pH 7, are lower compared to pH 9.
The equal sign is unnecessary in Fig.1B next to pH.
Lines 439-442, 456-460. The sentences need revision.
Section 3.1.4.
The comment about the importance of foaming properties is necessary. Is higher FC and FS favorable or unfavorable properties in putative applications of hydrolysates?
Section 3.2.
‘Chemical’ in the name of the section should be probably changed to ‘in vitro’ or omitted at all.
Section 3.2.1.
First sentence claim that DPPH radical scavenging activity was used to assess the antioxidant effect of protein hydrolysates in hydrophobic systems. Is the system used in this study hydrophobic?
Fig. 2 A-D should be individual pictures, not subpanels.
IC50 of DPPH radical scavenging activity of MW 10-30 and 5-10 kDa in Table 2 should be checked since it does not fit to the data in Fig. 2A.
Authors compare their data with Ref 3, however the concentration of DPPH used in Ref 3 is 0.1 mM whereas 0.1 M used by authors (probably a mistake). Therefore, the values cannot be compared.
Section 3.2.2.
Lines 549-553. Ref 3 and 10 use different reagents and concentrations for the assay therefore authors’ data cannot be compared directly with these Refs.
Lines 581-584. Incomplete sentence.
Section 3.2.4.
First sentence - emulsion of bulk lipid with low surface-to-volume ratios is more common in food and biological systems than what?
Last paragraph should be revised. Could authors prove that there is the synergistic effect not a simple combined effect of some amino acids?
Last sentence – greater potential than what?
Section 3.3.1.
Lines 710-714. It can not be stated that TPSPH and five fractions repaired cell damage induced by H2O2, because H2O2 was added afterwards.
Section 3.3.3.
First 3 sentences do not suit to start the section.
‘Normal group’ should be changed to ‘control group’.
Section 3.3.4.
The content MDA and LDH cannot be inhibited.
Section 3.3.5.
The concentration of H2O2 in this section do not correspond the concentration given in Methods section.
Conclusions
Speculations should be clearly separated from data-derived conclusions, i. e., the study do not give any result about the uptake of peptides by the cells, however, there is a sentence ‘Compared to the high molecular weight >30 kDa, the MW 5-10 kDa of TPSPH might enter efficiently cells and exhibit the highest well protective effect on HepG2 cells against oxidative stress induced by H2O2’.
Last sentence should be revised.
Author Response
A point-to-point response for revised manuscript # foods-1790850
Title: Antioxidant activity and functional properties of tree peony (Paeonia suffruticosa Andr.) seed protein hydrolysate and its cytoprotective effects on HepG2 cells
Authors: Ying-Qiu Li et al.
Submitted to: Foods.
The manuscript has been revised according to the comments of reviewers. A point-to-point response for revised manuscript was as followed:
Q1: The abstract contains a lot of abbreviations, most of them are clear for the readers from the field; however, HAAs and AAAs are both called antioxidant amino acids (line 26), what can be wrongly understood as full form of AAAs abbreviation.
A: According to the reviewer’s comments, “HAAs and AAAs” in the abstract were revised to “hydrophobic amino acids and aromatic amino acids”, respectively.
Q2: Sentences with ‘exhibited more effectively protect effects’ (line 27), and ‘activating the activities’ (line 29) should be changed.
A: According to the reviewer’s comments, sentences in line 27 and 29 were revised in the revised manuscript.
Q3: ‘…challenge problem…’ must be corrected in the first sentence.
A: According to the reviewer’s comments (Reviewer #2: I recommend a rewrite accompanied by a review of the text that should contain the ideas that the authors want the reader to capture when reading the manuscript), the introduction was rewritten and the sentence ‘…challenge problem…’ in the first sentence was deleted in the revised manuscript.
Q4: It would be clearer for the reader if the explanations of ROS effect in foods and body were not fused in every sentence of the first paragraph but explained separately.
A: According to the reviewer’s comments (Reviewer #2: I recommend a rewrite accompanied by a review of the text that should contain the ideas that the authors want the reader to capture when reading the manuscript), the introduction was rewritten and the first paragraph of the introduction was deleted in the revised manuscript.
Q5: Lines 62-67. The sentence needs a revision (fractionation…exhibited…activities, …scavenge of…).
A: According to the reviewer’s comments (Reviewer #2: I recommend a rewrite accompanied by a review of the text that should contain the ideas that the authors want the reader to capture when reading the manuscript), the introduction was rewritten and the sentence in Lines 62-67 was deleted in the revised manuscript.
Q6: Lines 68, 71. Abbreviations WSPHs-I and WSPHs are not given in full form, therefore it is difficult to understand the difference between them.
A: According to the reviewer’s comments, the full forms of WSPHs-I and WSPHs were presented in the revised manuscript.
Q7: Line 78. ‘…good bioavailable…’ must be corrected.
A: According to the reviewer’s comments (Reviewer #2: I recommend a rewrite accompanied by a review of the text that should contain the ideas that the authors want the reader to capture when reading the manuscript), the introduction was rewritten and the sentence ‘…good bioavailable…’ in line 78 was deleted in the revised manuscript.
Q8: Lines 96-99. Authors claim that ‘Tree peony (Paeonia suffruticosa Andr.) seed protein is a good source of protein with favourable amino acid profiles and desirable functional properties, such as emulsifying, foaming, water absorption and oil absorption properties’, but do not explain for which purpose these properties are favourable (as a food, food additive, preservative, drug/supplement etc.). Also, some data on the possible toxicity (or absence of toxicity) of tree peony seeds should be mentioned.
A: According to the reviewer’s comments (Reviewer #2: I recommend a rewrite accompanied by a review of the text that should contain the ideas that the authors want the reader to capture when reading the manuscript. For example, tree peony and its importance should be introduced from the beginning), the introduction was rewritten. And the data on the possible toxicity (or absence of toxicity) of tree peony seed protein were added in introduction.
Q9: Line 111. ‘…’TPSPH hydrolyzed with alcalase…’ must be corrected, since hydrolysate (TPSPH) wasn’t hydrolyzed again.
A: According to the reviewer’s comments, ‘TPSPH’ was revised to ‘tree peony seed protein’ in the revised manuscript.
Q10: The used sample solvent is missing in Sections 2.4.2, 2.5.1-4 .
A: According to the reviewer’s comments, the used sample solvent was supplemented in Sections 2.8.1, 2.9.1-4 in the revised manuscript.
Q11: Section 2.2. There is no information about mixing during hydrolysis procedure.
A: According to the reviewer’s comments, information on mixing during hydrolysis procedure was supplemented in Sections 2.2 in the revised manuscript.
Q12: Section 2.4.1. There is no explanation how the surface hydrophobicity is determined from fluorescence spectra.
A: According to the reviewer’s comments, the determination of surface hydrophobicity was interpreted in Section 2.4.1.
Q13: Section 2.4.2. There is ‘emulsion activity’ in the section title and ‘emulsifying activity’ in the text which is not the same. It is unclear how the pH was adjusted in the two-phase liquid system and what instrument was used for homogenization (centrifuge?). The equal sign must not be used instead of a dash in equation parameters explanation.
(1)There is ‘emulsion activity’ in the section title and ‘emulsifying activity’ in the text which is not the same.
A: According to the reviewer’s comments, the section titles in section 2.8.2 and Section 2.8.3 were revised to ‘emulsifying properties’ and ‘foaming properties’, respectively.
(2)It is unclear how the pH was adjusted in the two-phase liquid system and what instrument was used for homogenization (centrifuge?).
A: According to the reviewer’s comments, the pH was adjusted in the two-phase liquid system using 1.0 M HCl or 1.0 M NaOH. The homogeniser (FJ-200, Specimen Model Co., Shanghai, China) was used for homogenization. And these experimental conditions and instrument were supplemented in Section 2.8 in the revised manuscript.
(3) The equal sign must not be used instead of a dash in equation parameters explanation.
A: According to the reviewer’s comments, the equal sign in equation parameters explanation was deleted and these parameters were reinterpreted in the revised manuscript.
Q14: Section 2.4.3. The sentence in lines 213-214 is not full. It should be noted that initial volume V is before homogenization, and V0 and V30-after.
(1)The sentence in lines 213-214 is not full.
A: According to the reviewer’s comments, the sentence in lines 213-214 was revised in the revised manuscript.
(2)It should be noted that initial volume V is before homogenization, and V0 and V30-
A: According to the reviewer’s comments, the notes of initial volume V, V0 and V30 were supplemented in equation parameters explanation.
Q15: Section 2.5.3 Line 248. ‘…activity was conducted…’ must be corrected.
A: According to the reviewer’s comments, the sentense ‘The ferrous chelating activity was conducted by the method of…’ was revised to ‘The ferrous chelating activity was measured using the method of…’.
Q16: Section 2.5.4. Lines 278-280. What does ‘corrected absorbance’ mean?
A: According to the reviewer’s comments, ‘corrected absorbance’ was interpreted in the revised manuscript.
Q17: Section 2.6.1. Line 288. Cells were not subcultured in a 0.25% trypsin-EDTA solution but dissociated/detached using this solution.
A: According to the reviewer’s comments, the sentence in line 288 was revised to “The medium was refreshed every 2-3 days, and the cells at 80% to 90% confluence were subcultured and dissociated by using a 0.25% trypsin-EDTA solution.”
Q18:Section 2.6.2. A cell damage, induced by H2O2, cannot be called ‘stimulation’ (also further in the manuscript).
A: According to the reviewer’s comments, ‘stimulation’ in the manuscript was revised to ‘incubation’.
Q19: Section 2.6.3. Why was ROS generation measured in suspension but not in monolayer? How the resuspension procedure was performed (detachment solution, centrifugation, volumes used etc.)?
A: The cells were needed to wash twice with PBS buffer to remove extracellular DCFH-DA, so they were resuspended in PBS buffer.
Q20: Section 2.6.4.
Line 332. ‘…were added with…’ must be corrected.
The composition of lysis buffer is not given. Was it the same for all measurements described in the section?
The description of LDH leakage measurement is missing.
(1) Line 332. ‘…were added with…’ must be corrected.
A: According to the reviewer’s comments, the sentence ‘…were added with…’ was revised in the revised manuscript.
(2) The composition of lysis buffer is not given. Was it the same for all measurements described in the section? The description of LDH leakage measurement is missing.
A: According to the reviewer’s comments, the main compositions of lysis buffer were 20 mM Tris, 150 m M Na Cl, 1% Triton X-100, and 1 mM EDTA and they were supplemented in Section 2.10.4. It was the same for all measurements described in the section. The LDH leakage was measured by using an assay kit (Nanjing Jiancheng Bioengineering Institute, Nanjing, China).
Q21: Section 3.1.1.
The content of amino acids is given in percent in the text but in g/100 g dry basis in Table 1. The units should be unified.
A: According to the reviewer’s comments, the content of amino acids in the text was expressed in g/100 g dry basis.
Q22: Statistical significance marking is missing in the data of grouped amino acids in Table 1.
A: According to the reviewer’s comments, statistical significance marking of grouped amino acids was added in Table 1.
Q23: Section 3.1.2 Lines 384-385. ‘…fluorescence spectrum excited by binding ANS to hydrophobic groups…’ is not correct since the spectrum cannot be excited.
A: According to the reviewer’s comments, the sentense ‘…fluorescence spectrum excited by binding ANS to hydrophobic groups…’ in Lines 384-385 was revised to ‘ the surface hydrophobicity is positively related to the fluorescence intensity of ANS when binding to the proteins or hydrolysates’.
Q24: Line 388. ‘Fluorescence emission spectra…’ must be changed to ‘fluorescence spectra’.
A: According to the reviewer’s comments, ‘Fluorescence emission spectra…’ was revised to ‘fluorescence spectra’ in the the revised manuscript.
Q25: Line 388. The wavelength at which the fluorescence intensity was measured is not given.
A: According to the reviewer’s comments, the wavelengths at which the fluorescence intensity were added in the revised manuscript.
Q26: Fig. 1A y axis name is incorrect. Fig.1 A-E should be individual pictures, not subpanels.
A: According to the reviewer’s comments, y axis name of Fig. 1A was revised to “Fluorescence intensity (A.U.)”. And Fig.1 A-E were separated into individual pictures.
Q27: Lines 412-415. Authors claim, that TPSPH and five fractions at pH 3.0 and 5.0 exhibited lower EAI values, except for MW >30 kDa at pH 5.0, however, it is unclear compared to what? All EAI values, including pH 7, are lower compared to pH 9.
A: TPSPH and five fractions at pH 3.0 and 5.0 exhibited lower EAI values (7.63-19.09 m2/g) compared to their pH 7.0 and 9.0 (24.32-192.69 m2/g).
Q28: The equal sign is unnecessary in Fig.1B next to pH.
A: According to the reviewer’s comments, the equal sign in Fig.5A-D was deleted in the revised manuscript.
Q29: Lines 439-442, 456-460. The sentences need revision.
According to the reviewer’s comments, the sentences in Lines 439-442, 456-460 were revised in the revised manuscript.
Q30: The comment about the importance of foaming properties is necessary. Is higher FC and FS favorable or unfavorable properties in putative applications of hydrolysates?
A: According to the reviewer’s comments, the comment about the importance of foaming properties was added in the revised manuscript. Higher FC and FS are favorable or unfavorable properties depending on the environment in which the hydrolyzate is applied. In the present study, the high FC values of MW 10-30 kDa and MW 5-10 kDa in neutral and alkaline environments showed that they have a wider application in food processing such as ice creams, cakes and meringues.
Q31: Chemical’ in the name of the section should be probably changed to ‘in vitro’ or omitted at all.
A: According to the reviewer’s comments, Chemical’ in the name of the section was deleted in the revised manuscript.
Q32: First sentence claim that DPPH radical scavenging activity was used to assess the antioxidant effect of protein hydrolysates in hydrophobic systems. Is the system used in this study hydrophobic?
A: The system of DPPH radical scavenging activity used in this study is hydrophobic. DPPH is a hydrophobic radical with good solubility properties in low-polar solvents, so the DPPH radical scavenging activity assay can be used to evaluate the antioxidant activity of antioxidants in hydrophobic systems.
Q33: Fig. 2 A-D should be individual pictures, not subpanels.
A: Thanks for your kind comments, but we think that Fig. 2 A-D can not be individual pictures because there are too much figures in this manuscript.
Q34: IC50 of DPPH radical scavenging activity of MW 10-30 and 5-10 kDa in Table 2 should be checked since it does not fit to the data in Fig. 2A.
A: According to the reviewers’s comments, the IC50 values of DPPH radical scavenging activity of MW 10-30 and 5-10 kDa were revised in the revised manuscript.
Q35: Authors compare their data with Ref 3, however the concentration of DPPH used in Ref 3 is 0.1 mM whereas 0.1 M used by authors (probably a mistake). Therefore, the values cannot be compared.
A: According to the reviewers’s comments, the concentration of DPPH used in the section 2.9.1 was revised to 0.1 mM in the revised manuscript.
Q36: Lines 549-553. Ref 3 and 10 use different reagents and concentrations for the assay therefore authors’ data cannot be compared directly with these Refs.
A: According to the reviewers’s comments, the sentences about Ref 3 and 10 were deleted in the revised manuscript.
Q37: Lines 581-584. Incomplete sentence.
A: According to the reviewers’s comments, the setence was revised in the revised manuscript.
Q38: First sentence-emulsion of bulk lipid with low surface-to-volume ratios is more common in food and biological systems than what?
A: According to the reviewers’s comments, the first sentence was revised to “It has been recognized that high surface-to-volume ratios emulsions are more common than low surface-to-volume bulk lipid are more common in food and biological systems”.
Q39: Last paragraph should be revised. Could authors prove that there is the synergistic effect not a simple combined effect of some amino acids?
A: According to the reviewers’s comments, last paragraph was revised in the revised manuscript.
Q40: Last sentence–greater potential than what?
A: According to the reviewers’s comments, Last sentence–greater potential was revised to “great potential”.
Q41: Lines 710-714. It can not be stated that TPSPH and five fractions repaired cell damage induced by H2O2, because H2O2 was added afterwards.
A: According to the reviewers’s comments, the sentence in Lines 710-714 was revised to “This indicated TPSPH and five fractions reduced cell damage induced by H2O2”.
Q42: First 3 sentences do not suit to start the section.
A: According to the reviewers’s comments, the first 3 sentences was deleted in the revised manuscript.
Q43: ‘Normal group’ should be changed to ‘control group’.
A: According to the reviewers’s comments, ‘Normal group’ was revised to ‘control group’ in the revised manuscript.
Q44: The content MDA and LDH cannot be inhibited.
A: According to the reviewers’s comments, the sentence was revised to “the MDA and LDH contents were significantly reduced in TPSPH and five fractions treatment”.
Q45: The concentration of H2O2 in this section do not correspond the concentration given in Methods section.
A: According to the reviewers’s comments, the concentration of H2O2 in this section was revised to 500 μM.
Q46: Speculations should be clearly separated from data-derived conclusions, i. e., the study do not give any result about the uptake of peptides by the cells, however, there is a sentence ‘Compared to the high molecular weight >30 kDa, the MW 5-10 kDa of TPSPH might enter efficiently cells and exhibit the highest well protective effect on HepG2 cells against oxidative stress induced by H2O2’.
A: According to the reviewers’s comments, the sentence ‘ might enter efficiently cells’ was deleted in the revised manuscript.
Q47: Last sentence should be revised.
A: According to the reviewers’s comments, the last sentence was deleted in the revised manuscript.

Round 2
Reviewer 1 Report
According to my recommendation, the described peptide fractions were investigated by tricine SDS-electrophoresis. As can be seen from the presented data (Fig. 1), ultrafiltration of the protein hydrolysate resulted in 2 different fractions of peptides: with molecular weight > 12 kDa) (lane 2 in Fig. 1) and with molecular weight of 10–17 kDa (lane 3). By data of SDS-PAGE, the samples presented in lanes 3–6 contain the same set of peptides with decreasing concentration. No low-molecular peptides (< 10 kDa) were detected in the fractions presented in lanes 5 and 6. Thus, 4 of 5 studied fractions do not differ in the molecular weight of the peptides. The designation of the obtained fractions is misleading, since in fact this is not true. It remains unclear what is the difference between the fractions designated as MW<3 kDa, 3-5 kDa, 5-10 kDa and 10-30 kDa. It is possible that the investigated fractions differ from each other in the presence of low molecular weight components (amino acids, nucleotides, sugars, etc.), but this issue requires additional research and discussion.
Particle size distribution curves (Fig. 2) for fractions designated as MW <3 kDa, 3-5 and 5-10 kDa cannot be used to estimate particle sizes, because they look like random noise. In these experiments, several (8-10) measurements should be made, and the resulting statistical distribution should be presented as means ± SD. It is possible that these fractions contain only low molecular weight compounds that cannot be detected by this method. It should be noted that this method detects large aggregates of unknown nature, but not the size of the peptides.
Unfortunately, this study is not ready for publication.
Author Response
A point-to-point response for revised manuscript # foods-1790850R1
Title: Antioxidant activity, functional properties and cytoprotective effects on HepG2 cells of tree peony (Paeonia suffruticosa Andr.) seed protein hydrolysate as influenced by molecular weights fractionation
Authors: Ying-Qiu Li et al.
Submitted to: Foods.
Our responses to your questions and comments are as follows:
Q1: According to my recommendation, the described peptide fractions were investigated by tricine SDS-electrophoresis. As can be seen from the presented data (Fig. 1), ultrafiltration of the protein hydrolysate resulted in 2 different fractions of peptides: with molecular weight > 12 kDa) (lane 2 in Fig. 1) and with molecular weight of 10-17 kDa (lane 3). By data of SDS-PAGE, the samples presented in lanes 3-6 contain the same set of peptides with decreasing concentration. No low-molecular peptides (< 10 kDa) were detected in the fractions presented in lanes 5 and 6. Thus, 4 of 5 studied fractions do not differ in the molecular weight of the peptides. The designation of the obtained fractions is misleading, since in fact this is not true. It remains unclear what is the difference between the fractions designated as MW<3 kDa, 3-5 kDa, 5-10 kDa and 10-30 kDa. It is possible that the investigated fractions differ from each other in the presence of low molecular weight components (amino acids, nucleotides, sugars, etc.), but this issue requires additional research and discussion.
A: Thank you very much for the reviewer comments. According to the reviewer’s comments, the results about Non-reducing Tricine SDS-electrophoresis could not well indicate the molecular weight differences of TPSPH fractions. Therefore, the results and discussion of Non-reducing Tricine SDS-electrophoresis were deleted in the revised manuscript. In fact, just as the reviewer said, although the molecular weight bands in lanes 3-6 appeared in same lines, they might be composed of different components, such as amino acids, nucleotides and sugars. Tree peony seed protein was hydrolyzed by alcalase to obtain peptides of various molecular weights and free amino acids. These obtained peptides and amino acids after ultrafiltration might aggregate into fragments or grains by the action of chemical bonds. This aggregation phenomenon could affect the physicochemical properties and antioxidant activities of TPSPH fractions, which were evidenced by the results of amino acid compositions, surface hydrophobicity, functional properties, and antioxidant activity in our study. Therefore, the investigated fractions of this studies differ from each other. Just like what you said, this issue requires additional research and discussion. Thus the part of Tricine SDS-electrophoresis was deleted in the revised manuscript in consideration of no important effect on the integrity of the manuscript, and we will further investigate and discuss the question in follow-up study.
Q2: Particle size distribution curves (Fig. 2) for fractions designated as MW <3 kDa, 3-5 and 5-10 kDa cannot be used to estimate particle sizes, because they look like random noise. In these experiments, several (8-10) measurements should be made, and the resulting statistical distribution should be presented as means ± SD. It is possible that these fractions contain only low molecular weight compounds that cannot be detected by this method. It should be noted that this method detects large aggregates of unknown nature, but not the size of the peptides.
1) Particle size distribution curves (Fig. 2) for fractions designated as MW <3 kDa, 3-5 and 5-10 kDa cannot be used to estimate particle sizes, because they look like random noise. In these experiments, several (8-10) measurements should be made, and the resulting statistical distribution should be presented as means ± SD.
A: According to the reviewer’s comments, 10 measurements for particle size distribution of the TPSPH fractions were measured by using a laser particle-size analyzer (Zetasizer Nano ZS90, Malvern Instruments Co. Ltd., UK) according to dynamic light scattering and laser doppler. The resulting statistical distribution were presented as means ± SD. And the results and discussion of particle size distribution were re-described in the revised manuscript.
2) It is possible that these fractions contain only low molecular weight compounds that cannot be detected by this method. It should be noted that this method detects large aggregates of unknown nature, but not the size of the peptides.
A: In our study, we used the laser particle-size analyzer of Zetasizer Nano ZS90 (Malvern Instruments Co. Ltd., UK) to determine particle size distribution of the TPSPH fractions. The particle size test range of Zetasizer Nano ZS90 was 0.2 nm to 5.0 μm. It could not only detect large aggregates but the small particle size. The results showed that TPSPH fractions were characterized by a wide distribution of particle sizes between 50 nm and 7000 nm. Tian et al., (2020) employed the Zetasizer Nano ZS90 to determine the particle size of soybean protein hydrolysate, and they found that the particle size ranged from 40 to 100 nm. Chang et al., (2020) determined the particle size of potato protein isolate hydrolysate using a 90Plus Nanoparticle Size Analyzer, indicating that potato protein isolate hydrolysate had a particle size distribution of 0-50 nm. Xie et al., (2021) used Nano-ZSE particle size potentiometer to determine the particle sizes of rice dreg protein hydrolysates, indicating that rice dreg protein hydrolysates had the average particle size ranged from 165.4 nm to -295.3 nm. Ai et al., (2020) employed a B1-90PLus ano-meter analyzer to determine the particle size of preserved egg white hydrolysates obtained with different proteases and they found that the particle size ranged from 348.88 to 8316.58 nm.
Ai, M., Tang, T., Zhou, L., Ling, Z., Guo, S., & Jiang, A. Effects of different proteases on the emulsifying capacity, rheological and structure characteristics of preserved egg white hydrolysates. Food Hydrocolloids, 2019, 87, 933-942. https://doi.org/10.1016/j.foodhyd.2018.09.023.
Chang, C.Y., Jin, J.D., Chang, H.L., Huang, K.C., Chiang, Y.F. & Hsia, S.M. Physicochemical and Antioxidative Characteristics of Potato Protein Isolate Hydrolysate. Molecules, 2020, 25, 4450. https://doi.org/10.3390/molecules25194450.
Tian, R., Feng, J., Huang, G., Tian, B.O., Zhang, Y., Jiang, L., Sui, X. Ultrasound driven conformational and physicochemical changes of soy protein hydrolysates, Ultrasonics Sonochemistry, 2020, 68, 105202. https://doi.org/10.1016/j.ultsonch.2020.105202.
Xie, H.X. Huang, J.M., Woo, W.M. Hu, J.W., Xiong, H., Zhao, Q. Effect of cold and hot enzyme deactivation on the structural and functional properties of rice dreg protein hydrolysates. Food Chemistry, 2021, 345, 128784. https://doi.org/10.1016/j.foodchem.2020.128784.
